# GARNET: Reduced-Rank Topology Learning for Robust and Scalable Graph Neural Networks

**Chenhui Deng**
Cornell University
cd574@cornell.edu

**Xiuyu Li**
UC Berkeley
xiuyu@berkeley.edu

**Zhuo Feng**
Stevens Institute of Technology
zfeng12@stevens.edu

**Zhiru Zhang**
Cornell University
zhiruz@cornell.edu

## Abstract

Graph neural networks (GNNs) have been increasingly deployed in various applications that involve learning on non-Euclidean data. However, recent studies show that GNNs are vulnerable to graph adversarial attacks. Although there are several defense methods to improve GNN robustness by eliminating adversarial components, they may also impair the underlying clean graph structure that contributes to GNN training. In addition, few of those defense models can scale to large graphs due to their high computational complexity and memory usage. In this paper, we propose GARNET[1], a scalable spectral method to boost the adversarial robustness of GNN models. GARNET first leverages weighted spectral embedding to construct a base graph, which is not only resistant to adversarial attacks but also contains critical (clean) graph structure for GNN training. Next, GARNET further refines the base graph by pruning additional uncritical edges based on probabilistic graphical model. GARNET has been evaluated on various datasets, including a large graph with millions of nodes. Our extensive experiment results show that GARNET achieves adversarial accuracy improvement and runtime speedup over state-of-the-art GNN (defense) models by up to $10.23\%$ and $14.7\times$, respectively.

## 1 Introduction

Recent years have witnessed a surge of interest in graph neural networks (GNNs), which incorporate both graph structure and node attributes to produce low-dimensional embedding vectors that maximally preserve graph structural information [1]. GNNs have achieved promising results in various real-world applications, such as recommendation systems [2], self-driving car [3], and chip placements [4]. However, recent studies have shown that adversarial attacks on graph structure accomplished by inserting, deleting, or rewiring edges in an unnoticeable way, can easily fool the GNN models and drastically degrade their accuracy in downstream tasks (e.g., node classification) [5, 6].

In literature, one of the most effective ways to defend GNNs is to purify the graph by removing adversarial graph structures. Entezari et al. [7] observe that adversarial attacks mainly affect high-rank graph properties; thus they propose to first construct a low-rank graph by performing truncated singular value decomposition (TSVD) on the graph adjacency matrix, which can then be exploited for training a robust GNN model. Later, Jin et al. [8] propose Pro-GNN to jointly learn a new graph and a robust GNN model with the low-rank constraints imposed by the graph structure. While prior methods using low-rank approximation largely eliminate adversarial components in the graph spectrum, they involve dense adjacency matrices during GNN training, leading to a much higher time/space complexity and prohibiting their applications in large-scale graph learning tasks.

---

[1]Source code of GARNET is freely available at: github.com/cornell-zhang/GARNET.

C. Deng et al., GARNET: Reduced-Rank Topology Learning for Robust and Scalable Graph Neural Networks. *Proceedings of the First Learning on Graphs Conference (LoG 2022)*, PMLR 198, Virtual Event, December 9–12, 2022.

**Figure 1:** "TSVD AdvGraph" and "GARNET AdvGraph" denote adversarial graphs purified by TSVD and GARNET, respectively. (a) Graph rank comparison on Cora under Metattack with different perturbation ratio. (b) Singular value comparison of different normalized graph adjacency matrices on Cora. (c) Accuracy $\pm$ std. of GCN-TSVD on Cora with different $r$-rank approximation via TSVD.

In addition, due to the high computational cost of TSVD, existing low-rank based methods can only preserve top $r$ singular components (e.g., $r = 50$). Consequently, as shown in Figure 1(a), these methods may lose a wide range of clean graph spectrum that corresponds to important structures of the clean graph in the spatial domain. This is confirmed in Figure 1(c), where the clean accuracy of the TSVD-based method largely increases when preserving more spectral information via increasing the graph rank $r$. In other words, prior low-rank approximation methods eliminate high-rank adversarial components at the cost of inevitably impairing the important (clean) graph structure, which degrades the overall quality of the reconstructed graph and therefore limits the performance of GNN training.

In this work, we propose GARNET, a novel spectral approach to learning the underlying clean graph topology of an adversarial graph via combining spectral embedding with probabilistic graphical model (PGM), where the learned graph structure encodes the conditional dependence among low-dimensional node representations (spectral embedding vectors) [9]. More concretely, given an adversarial graph, GARNET first constructs a base graph topology by leveraging weighted spectral embeddings that are resistant to adversarial attacks, which is followed by an effective and efficient graph refinement scheme for pruning noncritical edges in the base graph by exploiting PGM.

By recovering the clean graph structure, Figures 1(a) and 1(b) show that the adversarial graph purified by GARNET largely restores the rank of the underlying clean graph. Thus, GARNET can be viewed as a reduced-rank topology learning approach that slightly reduces the rank of the input adversarial graph, which is fundamentally different from the prior low-rank based defense methods (e.g., TSVD and ProGNN). Moreover, GARNET scales comfortably to large graphs due to its nearly-linear algorithm complexity, and produces a sparse yet high-quality graph that improves GNN robustness without involving any dense adjacency matrices during GNN training. As a byproduct, unlike existing defense methods (e.g., ProGNN) that assume graphs to be homophilic, i.e, adjacent nodes in a graph tend to have similar attributes [10], GARNET does not have such an assumption and thus can protect GNNs against adversarial attacks on both homophilic and heterophilic graphs.

We evaluate GARNET on both homophilic and heterophilic datasets under strong graph adversarial attacks such as Nettack [5] and Metattack [6]. Moreover, we further show the nearly-linear scalability of our approach on the ogbn-products dataset that consists of millions of nodes [11]. Our experimental results indicate that GARNET largely improves both clean and adversarial accuracy over baselines in most cases. Our main technical contributions are summarized as follows:

• To our knowledge, we are the first to exploit spectral graph embedding and probabilistic graphical model for improving robustness of GNN models, which is achieved by learning a reduced-rank graph topology for recovering the underlying clean graph structure from the input adversarial graph.

• By recovering the critical edges that contribute to maximum likelihood estimation in PGM while ignoring adversarial components, GARNET produces a high-quality graph on which existing GNN models can be trained to achieve high accuracy. Our experimental results show that GARNET gains up to 10.23% adversarial accuracy improvement over state-of-the-art defense baselines.

• Our proposed reduced-rank topology learning method has a nearly-linear complexity in time/space and produces a sparse graph structure for scalable GNN training. This allows GARNET to run up to 14.7× faster than prior defense methods on popular data sets such as Cora and Squirrel. In addition, GARNET scales comfortably to very large graph data sets with millions of nodes, while prior defense methods run out of memory even on a graph with 20k nodes.

## 2 Background

### 2.1 Undirected Probabilistic Graphical Models

Consider an $n$-dimensional random vector $x$ that follows a multivariate Gaussian distribution $x \sim N(0, \Sigma)$, where $\Sigma = \mathbb{E}[xx^\top] \succ 0$ represents the covariance matrix, and $\Theta = \Sigma^{-1}$ represents the precision matrix (inverse covariance matrix). Given a data matrix $X \in R^{n \times d}$ that includes $d$ i.i.d (independent and identically distributed) samples $X = [x_1, ..., x_d]$, where $x_i \sim N(0, \Sigma)$ has an $n$-dimensional Gaussian distribution with zero mean, the goal of probabilistic graphical models (PGM) is to learn a precision matrix $\Theta$ that corresponds to an undirected graph structure $\mathcal{G}$ for encoding the conditional dependence between variables of the observations on columns of $X$ [12, 13]. Specifically, the classical graphical Lasso method aims at estimating a sparse $\Theta$ through maximum likelihood estimation (MLE) of $f(x)$ leveraging convex optimization [13]. In this work, we focus on one increasingly popular type of Gaussian graphical models, which is also known as attractive Gaussian Markov random fields (GMRFs). Attractive GMRFs restrict the precision matrix to be a Laplacian-like matrix $\Theta = L + \frac{I}{\sigma^2}$, where $L = D - A$ denotes the set of valid graph Laplacian matrices with $D$ and $A$ representing the diagonal degree matrix and adjacency matrix of the underlying undirected graph, respectively, $I$ denotes the identity matrix, and $\sigma^2$ is a constant denoting prior data variance. Similar to the graphical Lasso method [13], recent methods for estimating attractive GMRFs leverage emerging graph signal processing (GSP) techniques to solve the following convex problem [9, 14–17]:

$$\max_{\Theta} \log \det \Theta - \frac{1}{d} tr(XX^T \Theta) - \alpha \|\Theta\|_1 \tag{1}$$

where $det(\cdot)$ and $tr(\cdot)$ denote the determinant and trace operators, respectively, $\alpha$ is a hyperparameter to control the regularization term. The first two terms together can be interpreted as log-likelihood under a GMRF. The last $\ell_1$ regularization term is to enforce $\Theta$ (and the corresponding graph) to be sparse. If $X$ is non-Gaussian, Equation 1 can be regarded as Laplacian estimation based on minimizing the Bregman divergence between positive definite matrices induced by the function $\Theta \mapsto -\log \det(\Theta)$ [18].

### 2.2 Graph Adversarial Attacks

Most existing graph adversarial attacks aim at degrading the accuracy of GNN models by inserting/deleting edges in an unnoticeable way (e.g., maintaining node degree distribution) [19]. The most popular graph adversarial attacks fall into the following two categories: (1) targeted attack, (2) non-targeted attack. The targeted attacks attempt to mislead a GNN model to produce a wrong prediction on a target sample (e.g., node), while the non-targeted attacks strive to degrade the overall accuracy of a GNN model for the whole graph data set. Dai et al. [20] first formulate the targeted attack as a combinatorial optimization problem and leverages reinforcement learning to insert/delete edges such that the target node is misclassified. Zügner et al. [5] propose another targeted attack called Nettack, which produces an adversarial graph by maximizing the training loss of GNNs. Zügner and Günnemann [6] further introduce Metattack, a non-targeted attack that treats the graph as a hyperparameter and uses meta-gradients to perturb the graph structure. It is worth noting that graph adversarial attacks have two different settings: poison (perturb a graph prior to GNN training) and evasion (perturb a graph after GNN training). As shown by Zhu et al. [21], the poison setting is typically more challenging to defend, as it changes the graph structure that fools GNN training. Thus, we aim to improve model robustness against attacks under the poison setting.

### 2.3 Graph Adversarial Defenses

To defend GNN against adversarial attacks, Entezari et al. [7] first observe that Nettack, a strong targeted attack, only changes the high-rank information of the adjacency matrix. Thus, they propose to construct a low-rank graph by performing truncated SVD to undermine the effects of adversarial attacks. Later, Jin et al. [8] propose Pro-GNN that adopts a similar idea yet encourages nodes with similar attributes to be connected when jointly learning the low-rank graph and GNN model. Although those low-rank approximation based methods achieve state-of-the-art results on several datasets, they produce dense adjacency matrices that correspond to complete graphs, which would limit their applications for large graphs. Moreover, they only preserve a small region of the graph spectrum and thus may lose too much important information corresponding to the clean graph structure in the spatial

domain, which limits the performance of GNN training. Recently, Chang et al. [22] exploit Laplacian eigenpairs to guide GNN training, which produces a robust model with quadratic time complexity and is thus not scalable to large graphs. In addition to the aforementioned spectral-based defense methods, GCNJaccard [23] and RS-GNN [24] purify the adversarial graph by connecting nodes with similar attributes or same labels. However, those defense methods explicitly (or implicitly) assume the underlying graph to be homophilic, which results in rather poor performance when defending GNN models on heterophilic graphs. In contrast to the prior arts, GARNET achieves highly robust yet scalable performance on both homophilic and heterophilic graphs under adversarial attacks by leveraging a novel graph purification scheme based on spectral embedding and graphical model.

## 3 The GARNET Approach

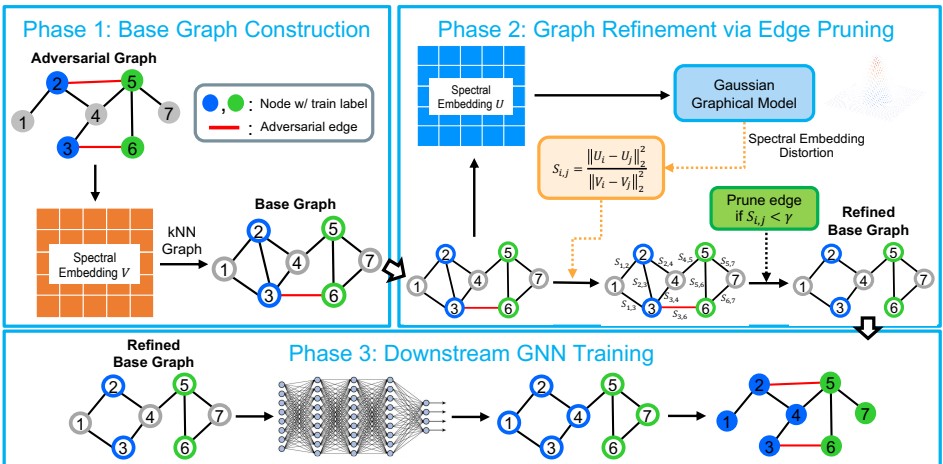

**Figure 2:** An overview of the three major phases of GARNET.

Recently, Entezari et al. [7] and Jin et al. [8] have shown that the well-known graph adversarial attacks (e.g., Nettack and Metattack) are essentially high-rank attacks, which increase graph rank by enlarging the smallest singular values of adjacency matrix when perturbing the graph structure, while rest of the graph spectrum remains almost the same. Consequently, a natural way for improving GNN robustness is to find the low-rank approximation of the adversarial adjacency matrix.

**Low-rank topology learning (prior work).** Given an adversarial adjacency matrix $A_{adv} \in R^{n \times n}$, Entezari et al. [7] propose to reconstruct a low-rank approximated adjacency matrix via performing TSVD: $\hat{A} = U\Sigma V^T$, where $\Sigma \in R^{r \times r}$ is a diagonal matrix consisting of $r$ largest singular values of $A_{adv}$. $U \in R^{n \times r}$ and $V \in R^{n \times r}$ contain the corresponding left and right singular vectors, respectively. As the largest singular values are hardly affected by graph adversarial attacks, the reconstructed low-rank adjacency matrix $\hat{A}$ is resistant to adversarial attacks.

However, due to the high computational cost of TSVD, $\hat{A}$ is typically computed by only using top $r$ largest singular values and their corresponding singular vectors, where $r$ is a relatively small number (e.g., $r = 50$). Consequently, the rank of $\hat{A}$ is only $r = 50$, which is two orders of magnitude smaller than the rank of the clean graph, as shown in Figure 1(a). Since these low-rank methods are overly aggressive in reducing the graph rank, $\hat{A}$ may lose too much important spectral information corresponding to the clean graph structure. As shown in Figure 1(c), the clean accuracy of the TSVD-based method is largely improved by increasing the graph rank $r$, which indicates the low-rank graph obtained with a small $r$ loses the key graph structure contributing to GNN training. Note that the adversarial and clean graphs share most of the graph structure, as adversarial attacks perturb the clean graph in an unnoticeable way. Consequently, losing those important clean graph structures will also limit the performance of GNN on the adversarial graph.

**Reduced-rank topology learning (this work).** Given the adversarial graph $\mathcal{G}_{adv}$ and its adjacency matrix $A_{adv}$, our goal is to learn a reduced-rank graph, which slightly reduces the rank of $\mathcal{G}_{adv}$ to mitigate the effects of adversarial attacks, while retaining most of the important graph spectrum

corresponding to the clean graph structure. As adversarial attacks mainly affect the least dominant singular components of $A_{adv}$ [7], one straightforward way for constructing such a reduced-rank graph is to utilize all the singular components except those least dominant ones via TSVD. Nonetheless, computing such a large number of singular components is computationally expensive [25], and is thus not scalable to large graphs.

To learn the reduced-rank graph in a scalable way, in this work, we leverage only the top few (e.g., 50) dominant singular components of $A_{adv}$ to restore its important graph spectrum, via recovering the corresponding clean graph structure with the aid of PGM. Figure 2 gives an overview of our proposed approach, GARNET, which consists of three major phases. The first phase constructs a base graph by exploiting spectral embedding and a scalable nearest-neighbor graph algorithm. The second phase further refines the base graph by pruning noncritical edges based on PGM. The last phase trains existing GNN models on the refined base graph to improve their robustness. Next, we will first describe our notion of clean graph recovery via PGM as well as the scalability issue of prior PGM-based work in Section 3.1, which motivates us to develop scalable GARNET kernels described in Sections 3.2 and 3.3. We further provide the overall complexity of GARNET in Section 3.4.

### 3.1 Graph Recovery via Graphical Model

A general philosophy behind PGM is that there exists an underlying graph $G$, whose structure determines the joint probability distribution of the observations on the data entities, i.e., columns of a data matrix $X \in R^{n \times d}$, where $n$ is the number of data points, $d$ the dimension per data point. To recover the underlying graph structure from the data matrix $X$, one common way is to leverage MLE by solving Equation 1 in Section 2.1. As the top few dominant singular components of the adjacency matrix capture the corresponding graph structure, we can naturally construct the data matrix $X$ based on those dominant singular components, and then adopt PGM to recover an underlying graph via MLE. To this end, we define a weighted spectral embedding matrix as follows:

**Definition 3.1.** Given the top $r$ smallest eigenvalues $\lambda_1, \lambda_2, ..., \lambda_r$ and their corresponding eigenvectors $v_1, v_2, ..., v_r$ of normalized graph Laplacian matrix $L_{norm} = I - D^{-\frac{1}{2}}AD^{-\frac{1}{2}}$, where $I$ and $A$ are the identity matrix and graph adjacency matrix, respectively, and $D$ is a diagonal matrix of node degrees, the **weighted spectral embedding matrix** is defined as $V \overset{\text{def}}{=} \left[ \sqrt{|1 - \lambda_1|}v_1, ..., \sqrt{|1 - \lambda_r|}v_r \right]$, whose $i$-th row $V_{i,:}$ is the **weighted spectral embedding** of the corresponding $i$-th node in the graph.

**Proposition 3.2.** *Given a normalized graph adjacency matrix $A_{norm} = D^{-\frac{1}{2}}AD^{-\frac{1}{2}}$ and weighted spectral embedding matrix $V$ of an undirected graph, let $\hat{A}$ be the rank-$r$ approximation of $A_{norm}$ via TSVD. If the top $r$ dominant eigenvalues of $A_{norm}$ are non-negative, then we have $\hat{A} = VV^T$.*

Our proof for Proposition 3.2 is available in Appendix A. Proposition 3.2 shows the connection between weighted spectral embedding and the low-rank adjacency matrix $\hat{A}$ obtained by TSVD. Specifically, the weighted spectral embedding matrix $V$ can be viewed as an eigensubspace matrix consisting of a few dominant singular components of the corresponding adjacency matrix. Thus, we can use $V$ to recover the underlying clean graph via PGM. However, obtaining $V$ requires the knowledge of the clean graph structure, which seems to create a chicken and egg problem.

Fortunately, since the dominant singular components are hardly affected by adversarial attacks [7], the weighted spectral embedding is therefore also resistant to adversarial attacks, indicating that the underlying clean graph $\mathcal{G}_{clean}$ and its corresponding adversarial graph $\mathcal{G}_{adv}$ share almost the same weighted spectral embeddings. As a result, we can exploit the weighted spectral embedding matrix $V$ of $\mathcal{G}_{adv}$ to represent that of $\mathcal{G}_{clean}$. By replacing the data matrix $X$ with $V$ in Equation 1, we have the following objective function:

$$\max_{\Theta} F := \log \det \Theta - \frac{1}{r} tr(VV^T\Theta) - \alpha \|\Theta\|_1 \tag{2}$$

More discussions on Equation 2 are available in Appendix Q. By finding the optimizer $\Theta^*$, we can recover the underlying graph that maximizes the likelihood given the observation on the weighted spectral embedding $V$. However, solving Equation 2 requires at least $O(n^2)$ time/space complexity per iteration with the most efficient algorithms, which thus cannot scale to large graphs [13, 26, 27].

As $\Theta$ is constrained to be a Laplacian-like matrix, finding the optimizer $\Theta^*$ in Equation 2 is equivalent to searching for critical edges from a complete graph, which would involve all possible (i.e., $O(n^2)$)

edges. Here we say an edge is critical (noncritical) if including it to the graph significantly increases (decreases) $F$ in Equation 2. Hence we can recover the underlying graph by pruning noncritical edges from the complete graph. However, storing a complete graph is still expensive. To have a near-linear algorithm for clean graph recovery, instead of searching in the complete graph, we limit our search within an initial base graph $\mathcal{G}_{base}$ that is much sparser but containing sufficient information for identifying the candidate edges critical to recover the clean graph. Subsequently, the final graphical model (graph Laplacian) can be obtained by further pruning noncritical edges from $\mathcal{G}_{base}$.

## 3.2 Base Graph Construction

During the first phase of GARNET (shown in Figure 2), our goal is to build a base graph $\mathcal{G}_{base}$, which greatly reduces the search space by not constructing a complete graph while preserving the critical candidate edges that are key to clean graph recovery. To this end, we give the following theorem:

**Theorem 3.3.** *Given a graph $\mathcal{G} = (\mathcal{V}, \mathcal{E})$ and its normalized Laplacian matrix $L_{\mathcal{G}}$, let $V_i$ denote the weighted spectral embedding of node $i$ by using top $r$ eigenpairs of $L_{\mathcal{G}}$. Suppose a relatively small $r$ is picked such that $\lambda_r \leq 1$, where $\lambda_r$ is the $r$-th smallest eigenvalue of $L_{\mathcal{G}}$, then we have $\sum_{(i,j)\in\mathcal{E}} \|V_i - V_j\|_2^2 \leq 0.25r$.*

Our proof for Theorem 3.3 is available in Appendix B. Note that $r$ is a small constant, which is independent of the graph size. Thus, Theorem 3.3 indicates that, if an edge connects nodes $i$ and $j$ in the clean graph, then the Euclidean distance between the weighted spectral embeddings of these two nodes will be small, which motivates us to build a k-nearest neighbor (kNN) graph as $\mathcal{G}_{base}$ to incorporate those clean edges.

Concretely, we first obtain the weighted spectral embedding matrix $V$ of the input adversarial graph $\mathcal{G}_{adv}$ to represent that of the underlying clean graph $\mathcal{G}_{clean}$, as $V$ consists of dominant singular components that are shared by $\mathcal{G}_{adv}$ and $\mathcal{G}_{clean}$ [7]. We then leverage $V$ to construct a kNN graph, where each node is connected to its $k$ most similar nodes based on the Euclidean distance between their spectral embeddings. Note that $V$ can be further concatenated with node feature matrix for constructing the kNN graph. A thorough discussion on incorporating node feature information is available in Appendix L. In this work, we exploit an approximate kNN algorithm for constructing the graph, which has $O(|\mathcal{V}| \log |\mathcal{V}|)$ complexity and thus can scale to very large graphs [28]. By choosing a proper $k$ (e.g., $k = 50$), $\mathcal{G}_{base}$ is likely to cover edges in the underlying clean graph. Thus, $\mathcal{G}_{base}$ can serve as a reasonable search space for identifying critical edges in the next step.

## 3.3 Graph Refinement via Edge Pruning

For the second phase of GARNET shown in Figure 2, we refine $G_{base}$ by aggressively pruning noncritical edges from $G_{base}$, such that the refined graph only preserves the most important edges that contribute most to the log-likelihood $F$ in Equation 2.

To identify critical (noncritical) edges that can most effectively increase (decrease) $F$, we exploit the update of $\Theta$ based on gradient ascent: $\Theta \leftarrow \Theta + \eta\frac{\partial F}{\partial\Theta}$, where $\eta$ is the step size. As mentioned in Section 2.1, $\Theta$ is constrained to be $L + \frac{I}{\sigma^2}$, which means the off-diagonal elements in $\Theta$ correspond to negative of edge weights in the underlying graph, i.e., $\Theta_{i,j} = -w_{i,j}$. Thus, the update of $\Theta_{i,j}$ during gradient ascent can be viewed as:

$$\Theta_{i,j} \leftarrow \Theta_{i,j} + \eta(\frac{\partial F}{\partial\Theta})_{i,j} = \Theta_{i,j} - \eta\frac{\partial F}{\partial w_{i,j}} \tag{3}$$

Equation 3 means that, if $\frac{\partial F}{\partial w_{i,j}}$ is large and positive, $\Theta_{i,j}$ will become more negative, which corresponds to increasing the edge weight in the underlying graph. Similarly, if $\frac{\partial F}{\partial w_{i,j}}$ is small and negative, $\Theta_{i,j}$ will be less negative, corresponding to decreasing the edge weight. In other words, the edge weight $w_{i,j}$ with a large (small) $\frac{\partial F}{\partial w_{i,j}}$ should be increased (decreased) to maximize the log-likelihood $F$, meaning the corresponding edge is critical (noncritical). Thus, we can identify the critical edges once we know $\frac{\partial F}{\partial w_{i,j}}$. By setting $\alpha = 0$ in Equation 2 (as GARNET naturally produces a sparse graph) and taking the partial derivative with respect to an edge weight $w_{i,j}$, we have:

$$\frac{\partial F}{\partial w_{i,j}} = \sum_{k=1}^{n} \frac{1}{\lambda_k + 1/\sigma^2} \frac{\partial \lambda_k}{\partial w_{i,j}} - \frac{\|V^T e_{i,j}\|_2^2}{r} \tag{4}$$

**Table 1:** Statistics of datasets used in our experiments.

| Dataset | Type | Homophily Score | Nodes | Edges | Classes | Features |
|---------|------|-----------------|-------|-------|---------|----------|
| Cora | Homophily | 0.80 | $2,485$ | $5,069$ | 7 | $1,433$ |
| Pubmed | Homophily | 0.80 | $19,717$ | $44,324$ | 3 | 500 |
| Chameleon | Heterophily | 0.23 | $2,277$ | $62,792$ | 5 | $2,325$ |
| Squirrel | Heterophily | 0.22 | $5,201$ | $396,846$ | 5 | $2,089$ |
| ogbn-arxiv | Homophily | 0.66 | $169,343$ | $1,166,243$ | 40 | 128 |
| ogbn-products | Homophily | 0.81 | $2,449,029$ | $61,859,140$ | 47 | 100 |

where $\lambda_k, \forall k = 1, 2, ..., n$ are the Laplacian eigenvalues of $\mathcal{G}_{base}$ (the initial graph for edge pruning), $e_{i,j} = e_i - e_j$, and $e_i$ denotes the vector with all zero entries except for the $i$-th entry being 1.

**Theorem 3.4** (Feng [17]). *Let $\lambda_k$ and $u_k$ be the $k$-th eigenvalue and the corresponding eigenvector of the Laplacian matrix, respectively. The spectral perturbation $\delta\lambda_k$ due to the increase of an edge weight $w_{i,j}$ can be estimated by $\delta\lambda_k = \delta w_{i,j}(u_k^T e_{i,j})^2$.*

The proof for Theorem 3.4 is available in Feng [17]. According to Theorem 3.4 and Equation 4, we can estimate $\frac{\partial F}{\partial w_{i,j}} \approx \|U^T e_{i,j}\|_2^2 - \frac{1}{r}\|V^T e_{i,j}\|_2^2$, where $U = [\frac{u_1}{\sqrt{\lambda_1 + 1/\sigma^2}}, ..., \frac{u_r}{\sqrt{\lambda_r + 1/\sigma^2}}]$, $\lambda_i$ is the $i$-th smallest Laplacian eigenvalue of $\mathcal{G}_{base}$, and $u_i$ is the corresponding eigenvector. Consequently, an edge $(i, j)$ is critical if $\|U^T e_{i,j}\|_2^2 \gg \frac{1}{r}\|V^T e_{i,j}\|_2^2$. As $V$ and $U$ are the spectral embeddings on the input adversarial graph and the base graph, respectively, we define the **spectral embedding distortion** $s_{i,j} = \frac{\|U^T e_{i,j}\|_2^2}{\|V^T e_{i,j}\|_2^2}$ to measure the edge importance. Consequently, we prune edges in the base graph $\mathcal{G}_{base}$ that have small spectral embedding distortion, i.e., $s_{i,j} < \gamma$, where $\gamma$ is a hyperparameter to control the sparsity of the refined graph. We further provide a strategy to simplify the distortion metric for edge pruning in Appendix S. Hence, the refined base graph $\mathcal{G}'_{base}$ largely recovers the underlying clean graph structure from the input adversarial graph. Since $\mathcal{G}'_{base}$ is constructed by only leveraging the top few dominant singular components of $\mathcal{G}_{adv}$, it ignores the high-rank adversarial components and thus robust to adversarial attacks. As a result, we can train a given GNN model on $\mathcal{G}'_{base}$ to improve its robustness, which is the last phase of GARNET.

### 3.4 Complexity of GARNET

The first phase of GARNET requires $O(r|\mathcal{E}|)$ time for computing top $r$ Laplacian eigenpairs [25], and $O(|\mathcal{V}| \log |\mathcal{V}|)$ time for kNN graph construction [28]. The second phase involves $O(rk|\mathcal{V}|)$ time for computing spectral embeddings and edge pruning on the kNN graph. Thus, the overall time complexity for graph purification is $O(r(|\mathcal{E}| + k|\mathcal{V}|) + |\mathcal{V}| \log |\mathcal{V}|)$, where $|\mathcal{V}|$ ($|\mathcal{E}|$) denotes the number of nodes (edges) in the adversarial graph, and $k$ is the averaged node degree in the kNN graph. Our systematic approach of choosing $r$ and the space complexity analysis are in Appendix F.

## 4 Experiments

We have conducted comparative evaluation of GARNET against state-of-the-art defense GNN models under targeted attack (Nettack) [5] and non-targeted attack (Metattack) [6] on both homophilic and heterophilic datasets. Besides, we also evaluate GARNET robustness against adaptive attacks. In addition, we further show the scalability of GARNET by comparing its run time with prior defense methods and evaluating GARNET on ogbn-products, which consists of more than 2 million nodes [11]. Finally, we conduct ablation studies to understand the effectiveness of GARNET kernels.

**Experimental Setup.** Table 1 shows the statistics of the datasets used in our experiments. We follow Zhu et al. [10] to compute the homophily score per dataset (lower score means more heterophilic). More details of datasets are available in Appendix C. We choose as baselines two state-of-the-art defense methods based on graph purification: TSVD [7] and Pro-GNN [8]. Besides, we evaluate training based defense methods GCN-LFR [22] and GNNGuard [29] on homophilic and heterophilic graphs, respectively. Moreover, we use GCN [30] and GPRGNN [31] as the backbone GNN models for defense on homophilic datasets (i.e., Cora and Pubmed). As GCN performs poorly on heterophilic datasets [10, 32], we choose GPRGNN as the backbone model on Chameleon and Squirrel datasets. Due to the space limit, we provide defense results with H2GCN [10] as the backbone model in Appendix J. For all baselines, we tune their hyperparameters against adversarial attacks with a

**Table 2:** Averaged node classification accuracy (%) $\pm$ std under targeted attack (Nettack) and non-targeted attack (Metattack) on homophilic graphs — We bold and underline the first and second highest accuracy of each backbone GNN model, respectively. $OOM$ means out of memory.

| Model | Cora (Nettack) | | Cora (Metattack) | | Pubmed (Nettack) | | Pubmed (Metattack) | |
|---|---|---|---|---|---|---|---|---|
| | Clean | Adversarial | Clean | Adversarial | Clean | Adversarial | Clean | Adversarial |
| GCN-Vanilla | $\underline{80.96} \pm 0.95$ | $55.66 \pm 1.95$ | $\mathbf{81.35} \pm 0.66$ | $56.28 \pm 1.19$ | $87.26 \pm 0.51$ | $66.67 \pm 1.34$ | $\mathbf{87.16} \pm 0.09$ | $77.20 \pm 0.27$ |
| GCN-TSVD | $72.65 \pm 2.29$ | $60.30 \pm 2.25$ | $73.86 \pm 0.53$ | $62.44 \pm 1.16$ | $87.03 \pm 0.48$ | $\underline{79.56} \pm 0.48$ | $84.53 \pm 0.08$ | $\underline{84.30} \pm 0.08$ |
| GCN-ProGNN | $80.54 \pm 1.21$ | $\underline{65.38} \pm 1.65$ | $78.56 \pm 0.36$ | $\underline{72.28} \pm 1.67$ | $\mathbf{88.14} \pm 1.44$ | $71.89 \pm 1.56$ | $84.62 \pm 0.11$ | $83.89 \pm 0.32$ |
| GCN-LFR | $80.07 \pm 0.95$ | $53.73 \pm 2.17$ | $77.23 \pm 2.61$ | $65.38 \pm 3.71$ | $87.20 \pm 1.24$ | $68.49 \pm 2.44$ | $81.91 \pm 0.26$ | $78.32 \pm 0.69$ |
| GCN-GARNET | $\mathbf{81.08} \pm 2.05$ | $\mathbf{67.04} \pm 2.05$ | $\underline{79.64} \pm 0.75$ | $\mathbf{73.89} \pm 0.91$ | $\underline{87.96} \pm 0.58$ | $\mathbf{86.12} \pm 0.86$ | $\underline{85.37} \pm 0.20$ | $\mathbf{85.14} \pm 0.23$ |
| GPR-Vanilla | $\mathbf{83.04} \pm 2.05$ | $62.89 \pm 1.95$ | $\mathbf{83.05} \pm 0.42$ | $74.27 \pm 2.11$ | $\underline{90.05} \pm 0.73$ | $\underline{76.99} \pm 1.16$ | $\mathbf{87.35} \pm 0.13$ | $\underline{84.18} \pm 0.15$ |
| GPR-TSVD | $81.68 \pm 1.78$ | $63.52 \pm 3.27$ | $81.61 \pm 0.54$ | $\underline{78.50} \pm 1.20$ | $OOM$ | $OOM$ | $OOM$ | $OOM$ |
| GPR-ProGNN | $82.04 \pm 1.33$ | $\underline{63.74} \pm 2.57$ | $82.04 \pm 0.90$ | $76.29 \pm 1.46$ | $OOM$ | $OOM$ | $OOM$ | $OOM$ |
| GPR-GARNET | $\underline{82.77} \pm 1.89$ | $\mathbf{71.45} \pm 2.73$ | $\underline{82.67} \pm 1.89$ | $\mathbf{81.34} \pm 0.79$ | $\mathbf{90.99} \pm 0.52$ | $\mathbf{89.52} \pm 0.45$ | $\underline{86.86} \pm 0.57$ | $\mathbf{85.69} \pm 0.26$ |

**Table 3:** Averaged node classification accuracy (%) $\pm$ std on heterophilic graphs — We bold and underline the first and second highest accuracy, respectively. The backbone GNN model is GPRGNN.

| Model | Chameleon (Nettack) | | Chameleon (Metattack) | | Squirrel (Nettack) | | Squirrel (Metattack) | |
|---|---|---|---|---|---|---|---|---|
| | Clean | Adversarial | Clean | Adversarial | Clean | Adversarial | Clean | Adversarial |
| Vanilla | $71.46 \pm 1.92$ | $\underline{66.26} \pm 1.71$ | $\mathbf{61.36} \pm 1.00$ | $\underline{53.20} \pm 0.88$ | $\underline{41.36} \pm 2.87$ | $\underline{39.45} \pm 2.36$ | $39.51 \pm 1.64$ | $\underline{35.22} \pm 1.20$ |
| TSVD | $62.12 \pm 3.04$ | $60.37 \pm 2.86$ | $47.29 \pm 1.63$ | $45.12 \pm 1.34$ | $32.98 \pm 2.36$ | $31.20 \pm 1.84$ | $31.36 \pm 1.87$ | $23.91 \pm 1.40$ |
| ProGNN | $58.80 \pm 1.72$ | $57.07 \pm 1.82$ | $48.39 \pm 0.68$ | $46.69 \pm 0.61$ | $31.81 \pm 1.72$ | $27.27 \pm 1.87$ | $31.64 \pm 2.87$ | $29.36 \pm 3.61$ |
| GNNGuard | $64.87 \pm 2.62$ | $62.21 \pm 1.94$ | $58.01 \pm 1.57$ | $49.89 \pm 1.34$ | $34.17 \pm 2.33$ | $33.41 \pm 1.82$ | $37.46 \pm 0.56$ | $32.69 \pm 0.59$ |
| GARNET | $\mathbf{72.89} \pm 2.65$ | $\mathbf{71.83} \pm 2.11$ | $\underline{61.11} \pm 2.46$ | $\mathbf{59.96} \pm 0.84$ | $\mathbf{44.91} \pm 1.53$ | $\mathbf{43.64} \pm 1.53$ | $\mathbf{43.43} \pm 1.14$ | $\mathbf{41.97} \pm 1.02$ |

small perturbation, and keep the same hyperparameters for larger adversarial perturbations. Detailed hyperparameter settings of baselines and GARNET are available in Appendix D. Our hardware information is provided in Appendix E.

## 4.1 Robustness of GARNET

**Defense on homophilic graphs**. We first evaluate the model robustness on homophilic graphs against the targeted attack (Nettack) and the non-targeted attack (Metattack). Specifically, Nettack aims to fool a GNN model to misclassify some target nodes with a few structure (edge) perturbations. The goal of Metattack is to drop the overall accuracy of the whole test set with a given perturbation ratio budget (i.e., the number of adversarial edges over the number of total edges). Due to the space limit, we only show defense results under Nettack and Metattack with 5 perturbed edges per target node and 20% perturbation ratio, respectively. Results with other perturbation budgets are in Appendix I.

Table 2 reports the average accuracy over 10 runs on Cora and Pubmed. It shows that GARNET, with either a backbone GNN model (GCN or GPRGNN), outperforms defense baselines in terms of both clean and adversarial accuracy in most cases. We attribute the large accuracy improvement to GARNET's strengths in recovering key structures of the clean graph while ignoring the high-rank adversarial components during graph purification. Moreover, as both TSVD and ProGNN involve dense matrices during GNN training, they run out of GPU memory even on Pubmed, a graph with only 20k nodes. In contrast, GARNET is not only robust to adversarial attacks, but also scalable to large graphs, as empirically shown in Section 4.2.

**Defense on heterophilic graphs**. We report the averaged accuracy over 10 runs on heterophilic graphs in Table 3, which shows that all defense baselines fail to defend GPRGNN on heterophilic graphs and even degrade the accuracy of the vanilla GPRGNN by a large margin. The reason why ProGNN performs poorly is that it follows the graph homophily assumption for improving GNN robustness, which contradicts the property of heterophilic graphs. For the TSVD-based defense method, the low-rank graph generated by TSVD contains negative edge weights, which degrade the performance of GPRGNN for adapting its graph filter on heterophilic graphs [31]. Although Zhang and Zitnik [29] have shown GNNGuard can improve model robustness on synthetic heterophillic graphs, our results indicate that it fails to defend GNN models on realistic heterohilic graphs. We attribute it to that the quality of graphlet degree vectors used in GNNGuard is degraded by structural perturbations induced via adversarial attacks. In contrast, GARNET largely recovers the clean graph structure based on Theorem 3.3 without the assumption on whether adjacent nodes have similar

**Table 4:** Averaged accuracy (%) $\pm$ std on Cora under Metattack and LowBlow with 20% perturbation ratio. We use GPRGNN as the backbone GNN model.

| Model | Metattack | LowBlow |
|---|---|---|
| Vanilla | $74.27 \pm 2.11$ | $74.77 \pm 0.71$ |
| TSVD | $78.50 \pm 1.20$ | $26.03 \pm 2.76$ |
| ProGNN | $76.29 \pm 1.46$ | $69.88 \pm 1.61$ |
| GARNET | $\mathbf{81.34} \pm 0.79$ | $\mathbf{77.71} \pm 0.95$ |

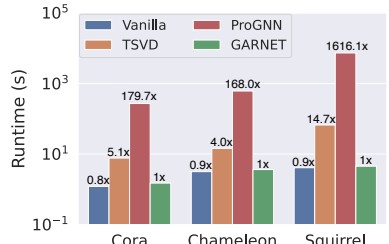

**Figure 3:** End-to-end runtime comparison of GARNET and baseline methods.

attributes. In other words, GARNET will produce a heterophilic graph if the underlying clean graph is heterophilic, which is further confirmed in Appendix O. Consequently, GARNET improves accuracy over defense baselines by up to 10.23% (i.e., 43.64% − 33.41% on Squirrel under Nettack) on heterophilic graphs.

**Defense against adaptive attacks.** As GARNET is non-differentiable during kNN graph construction, it is difficult to optimize a specific loss function for adaptive attack. Instead, we adopt an attack called LowBlow from [7] (based on Metattack), which deliberately perturbs low-rank singular components in the graph spectrum, yet violates the unnoticeable condition (i.e., preserving node degree distribution after attacking). Since LowBlow has cubic complexity for computing the full set of adjacency eigenpairs, we only show results on the small graph Cora in Table 4, which indicates GARNET still achieves the highest adversarial accuracy under LowBlow, while all low-rank defense baselines perform even worse than vanilla GPRGNN model. The reason lies in that the kNN graph (with a relatively large $k$) in GARNET is less vulnerable to the perturbations of weighted spectral embeddings (i.e., low-rank singular components) [33], compared to prior low-rank defense methods.

## 4.2 Scalability of GARNET

**Table 5:** Averaged accuracy (%) $\pm$ std under GR-BCD attack.

| | ogbn-arxiv | | | ogbn-products | | |
|---|---|---|---|---|---|---|
| Model | Clean | 25% Ptb. | 50% Ptb. | Clean | 25% Ptb. | 50% Ptb. |
| GCN | $\mathbf{70.74} \pm 0.26$ | $45.18 \pm 0.25$ | $39.12 \pm 0.27$ | $\underline{75.68} \pm 0.20$ | $64.70 \pm 0.43$ | $62.71 \pm 0.44$ |
| GNNGuard | $68.78 \pm 0.32$ | $\underline{47.46} \pm 0.11$ | $\underline{41.18} \pm 0.12$ | $74.82 \pm 0.11$ | $\underline{66.76} \pm 0.23$ | $\underline{63.22} \pm 0.26$ |
| GCNJaccard | $67.77 \pm 0.18$ | $46.27 \pm 0.11$ | $40.84 \pm 0.19$ | $72.95 \pm 0.08$ | $60.90 \pm 0.18$ | $58.84 \pm 0.20$ |
| Soft Median GDC | $69.75 \pm 0.03$ | $45.31 \pm 0.06$ | $40.11 \pm 0.06$ | $66.31 \pm 0.03$ | $60.59 \pm 0.05$ | $59.73 \pm 0.05$ |
| GARNET | $\underline{69.91} \pm 0.29$ | $\mathbf{61.32} \pm 0.20$ | $\mathbf{60.88} \pm 0.13$ | $\mathbf{76.05} \pm 0.19$ | $\mathbf{75.03} \pm 0.14$ | $\mathbf{74.97} \pm 0.24$ |

To demonstrate the scalability of GARNET, we first compare the run time of GARNET with prior low-rank defense methods with GPRGNN as the backbone GNN model. As shown in Figure 3, the TSVD defense method is slower than GARNET since it produces a dense adjacency matrix that slows down the GNN training. Moreover, ProGNN is extremely slow as it jointly learns the low-rank graph structure and the robust GNN model, which requires performing TSVD for every epoch. In contrast, GARNET can efficiently produce a sparse graph for downstream GNN training, leading to end-to-end runtime speedup over prior methods by up to $14.7\times$.

In addition, we further evaluate the robustness of GARNET on two large datasets: ogbn-arxiv and ogbn-products, under powerful and scalable attacks proposed by [34]. As we run out of GPU memory when performing the PR-BCD attack, we choose the more scalable version GR-BCD that has less memory usage. We use GCN as the backbone model since it outperforms GPRGNN on large graphs. As TSVD and ProGNN run out of memory on these two datasets, we choose GNNGuard, GCNJaccard [23], and Soft Median GDC [34] as baselines. Table 5 shows GARNET achieves comparable clean accuracy compared to GCN, and drastically improves the adversarial accuracy over defense baselines by up to 16.13%. Moreover, we also evaluate the run time of GARNET on the large graphs. Concretely, the end-to-end run time of GARNET is 40 mins and 4 hours on ogbn-arxiv and ogbn-products, respectively, which is $3\times$ faster than the most competitive baseline GNNGuard that takes more than 2 hours on ogbn-arxiv and 11 hours on ogbn-products. We provide potential ways to further accelerate GARNET in Appendix N.

### 4.3 Ablation Analysis of GARNET

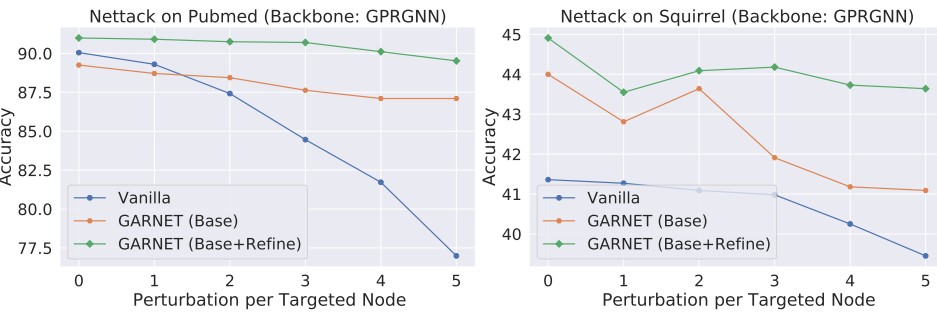

**Figure 4:** Ablation study of GARNET on graph refinement.

Figure 4 shows the comparison of GARNET results with and without graph refinement. When only constructing the base graph, GARNET achieves better adversarial accuracy than the vanilla GNN model, which confirms our Theorem 3.3 that the base graph construction can successfully recover clean graph edges. The graph refinement step further improves GARNET accuracy ($\sim 2\%$ increase) since some noncritical or even harmful edges are removed based on PGM. Due to the space limitation, the ablation studies of GARNET on the kNN graph and edge pruning are available in Appendix G.

### 4.4 Visualization

We visualize the local structure (within 2-hop neighbors) of a target node (randomly picked) on Cora in Figure 5. By comparing Figures 5(b) and 5(c), it is clear that GARNET effectively removes most of the adversarial edges induced by Nettack that connect nodes with different labels [8]. As a result, it is trivial for the backbone GNN model to correctly predict the target node since the surrounding nodes share the same label as the target node in GARNET graph. This explains why GARNET substantially improves the adversarial accuracy of GNN models. More visualizations are available in Appendix M.

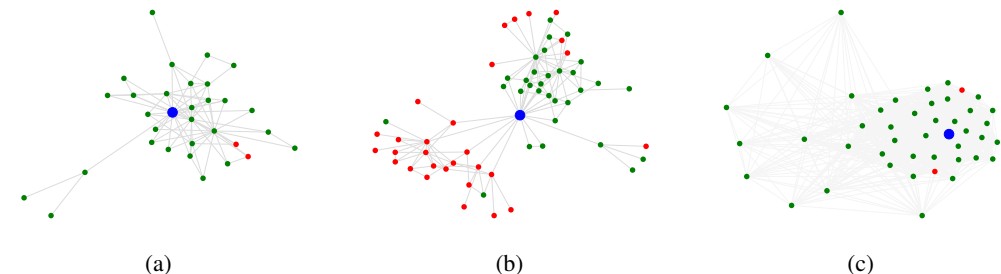

(a)                          (b)                          (c)

**Figure 5:** Visualizations on the same target node (marked in blue) as well as its 1-hop and 2-hop neighbors. Neighbor nodes are marked in green if they have the same label as the target node, and red otherwise. (a) clean graph. (b) adversarial graph. (c) adversarial graph purified by GARNET.

## 5 Conclusions

This work introduces GARNET, a spectral approach to robust and scalable graph neural networks by combining spectral embedding and the probabilistic graphical model. GARNET first uses weighted spectral embedding to construct a base graph, which is then refined by pruning uncritical edges based on the graphical model. Results show that GARNET not only outperforms state-of-the-art defense models, but also scales to large graphs with millions of nodes. An interesting direction for future work is to incorporate high-order structural information (e.g., motifs) to further boost model robustness.

## Acknowledgements

This work is supported in part by NSF grants #2021309, #2205572, #2212370, and #2212371, and a Facebook Research Award.

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

## A  Proof for Proposition 3.2

*Proof.* As the graph is undirected, we can perform eigendecomposition on both $A_{norm}$ and $L_{norm}$ to obtain their real eigenvalues and the corresponding eigenvectors. Let $\lambda_i$, $\hat{\lambda}_i$, and $\sigma_i$, $i = 1, 2, ..., r$ denote the $r$ smallest eigenvalues of $L_{norm}$, $r$ largest eigenvalues of $A_{norm}$, and $r$ largest singular values of $A_{norm}$, respectively. Since $A_{norm} = I - L_{norm}$, $A_{norm}$ and $L_{norm}$ share the same set of eigenvectors while their eigenvalues satisfy: $\hat{\lambda}_i = 1 - \lambda_i$, $i = 1, 2, ..., r$. Moreover, since we assume that the $r$ largest magnitude eigenvalues of $A_{norm}$ are non-negative, we have $\sigma_i = \left|\hat{\lambda}_i\right| = \hat{\lambda}_i$, $i = 1, 2, ..., r$. Thus, we have:

$$
\begin{aligned}
VV^T &= [v_1, ..., v_r]
\begin{bmatrix}
|1 - \lambda_1| & & \\
& \ddots & \\
& & |1 - \lambda_r|
\end{bmatrix}
[v_1, ..., v_r]^T \\
&= [v_1, ..., v_r]
\begin{bmatrix}
\left|\hat{\lambda}_1\right| & & \\
& \ddots & \\
& & \left|\hat{\lambda}_r\right|
\end{bmatrix}
[v_1, ..., v_r]^T \\
&= [v_1, ..., v_r]
\begin{bmatrix}
\sigma_1 & & \\
& \ddots & \\
& & \sigma_r
\end{bmatrix}
[v_1, ..., v_r]^T \\
&= \hat{A}
\end{aligned}
$$

$\square$

## B  Proof for Theorem 3.3

*Proof.* Since the weighted embedding matrix $V$ is defined as $V \overset{\text{def}}{=} \left[\sqrt{|1 - \lambda_1|}v_1, ..., \sqrt{|1 - \lambda_r|}v_r\right]$, where $\lambda_1, \lambda_2, ..., \lambda_r$ and $v_1, v_2, ..., v_r$ are the top $r$ smallest eigenvalues and the corresponding eigenvectors of normalized graph Laplacian matrix $L_{norm} = I - D^{-\frac{1}{2}}AD^{-\frac{1}{2}}$, we have:

$$
\begin{aligned}
\sum_{(i,j)\in\mathcal{E}} \|V_i - V_j\|_2^2 &= \sum_{k=1}^{r}\sum_{(i,j)\in\mathcal{E}} |1 - \lambda_k|\, (v_{k,i} - v_{k,j})^2 \\
&= \sum_{k=1}^{r} |1 - \lambda_k| \sum_{(i,j)\in\mathcal{E}} (v_{k,i} - v_{k,j})^2 \\
&= \sum_{k=1}^{r} |1 - \lambda_k|\, v_k^T L_{norm} v_k \\
&= \sum_{k=1}^{r} (1 - \lambda_k)\lambda_k \\
&\leq \sum_{k=1}^{r} 0.25 \\
&= 0.25r
\end{aligned}
$$

$\square$

The fourth equation above is based on Courant-Fischer Theorem [35] with the assumption that $\lambda_r \leq 1$ and the Laplacian eigenvectors are normalized (i.e., $\|v_k\|_2 = 1, \forall k = 1, ..., r$). The inequality is derived by arithmetic mean-geometric mean (AM-GM) inequality.

## C   Dataset Details

As in Jin et al. [8], we extract the largest connected components of the original Cora and Pubmed datasets [36] for the adversarial evaluation, with the same train/validation/test split. For Chameleon and Squirrel [37], we keep the same split setting as Chien et al. [31]. Finally, we follow the split setting of Open Graph Benchmark (OGB) [11] on ogbn-arxiv and ogbn-products. Note that all data used in our experiments do not contain personally identifiable information or offensive content.

In addition, we follow Jin et al. [8] for the selection of target nodes on Cora and Pubmed under Nettack. For the Chameleon and Squirrel datasets under Nettack, we choose target nodes that have degrees within the range of $[20, 50]$ and $[20, 140]$, respectively. In regard to non-targeted attacks (i.e., Metattack), we choose nodes in the test set as target nodes for all datasets. We implement all the adversarial attacks based on the DeepRobust library [38].

## D   Hyperparameters Settings

### D.1   Backbone GNN Models

**GCN.** We choose the GCN hyperparameters based on the DeepRobust library [38].

**GPRGNN.** We follow the hyperparameter settings provided at github.com/jianhao2016/GPRGNN with slightly different dropout rates (chosen from $0.3, 0.5, 0.7$) and learning rates (chosen from $0.01, 0.05, 0.1$). Specifically, we provide the complete choices of dropout rates and learning rates across all datasets and attack settings below:

- Cora-Nettack: dropout of $0.5$ and learning rate of $0.01$.
- Cora-Metattack: dropout of $0.5$ and learning rate of $0.01$.
- Pubmed-Nettack: dropout of $0.5$ and learning rate of $0.01$.
- Pubmed-Metattack: dropout of $0.5$ and learning rate of $0.01$.
- Chameleon-Nettack: dropout of $0.5$ and learning rate of $0.05$.
- Chameleon-Metattack: dropout of $0.3$ and learning rate of $0.05$.
- Squirrel-Nettack: dropout of $0.5$ and learning rate of $0.1$.
- Squirrel-Metattack: dropout of $0.5$ and learning rate of $0.1$.

**H2GCN.** We train a three-layer model in full batch, with a learning rate of $0.01$, dropout of $0.5$, hidden dimension of $64$, and $300$ epochs for both Chameleon and Squirrel datasets.

### D.2   Defense Baselines

**TSVD.** We use the same r eigenvectors in TSVD as those used in GARNET, which is shown in Table 6.

**GCNJaccard.** We choose the GCNJaccard hyperparameters based on the DeepRobust library [38].

**GNNGuard.** We set edge pruning threshold (the only hyperparameter in GNNGuard) to be $P_0 = 0.1$.

**Soft Median GDC.** We strictly follow the hyperparameter setting suggested by Geisler et al. [34]. In particular, we choose temperature $T$ of $5.0$ for soft median, $\alpha$ of $0.1$ ($0.15$) and $k$ of $64$ ($32$) for GDC on ogbn-arxiv (ogbn-products).

**ProGNN.** We find out its performance is very sensitive to hyperparameters. Thus we strictly follow the tuned hyperparameters available at github.com/ChandlerBang/Pro-GNN/scripts. As GCN-ProGNN training is very slow on Pubmed (estimated time is 30 days for 10 runs), we follow the suggestion from ProGNN authors to replace "svd" with "truncated svd" in the ProGNN implementation.

### D.3   GARNET

We run all GNN training with a full batch way and show the hyperparameters of GARNET on different datasets under Nettack (1 perturbation per node), Metattack ($10\%$ perturbation ratio), and GR-BCD ($25\%$ perturbation ratio) in Table 6. Note that we provide our strategy of choosing $r$ in Appendix F,

**Table 6:** Summary of hyperparameters in GARNET— We denote the number of eigenpairs for spectral embedding and nearest neighbors for base graph construction by $r$ and $k$, respectively.

| Dataset | $r$ | $k$ |
|---|---|---|
| Cora-Nettack | 50 | 30 |
| Cora-Metattack | 50 | 30 |
| Pubmed-Nettack | 50 | 50 |
| Pubmed-Metattack | 50 | 50 |
| Chameleon-Nettack | 50 | 50 |
| Chameleon-Metattack | 50 | 50 |
| Squirrel-Nettack | 50 | 50 |
| Squirrel-Metattack | 50 | 50 |
| ogbn-arxiv-GRBCD | 500 | 50 |
| ogbn-products-GRBCD | 500 | 50 |

which avoids conducting hyperparameter tuning on $r$ per dataset. Besides, we set the prior data variance $\sigma^2$ to be positive infinity for all graphs (i.e., ignore self-loops of $\Theta$ in Equation 2). Last but not least, we refer readers to our official configuration files at github.com/cornell-zhang/GARNET/configs for the detailed choices of the threshold $\gamma$ for edge pruning.

## E   Hardware Information

We conduct all experiments on a Linux machine with an Intel Xeon Gold 5218 CPU (8 cores @ 2.30GHz), 8 NVIDIA RTX 2080 Ti GPU (11 GB memory per GPU), and 1 RTX A6000 GPU (48 GB memory).

## F   Complexity Analysis of GARNET

### F.1   Time Complexity – Choice of $r$

We choose $r$ based on the number of classes per dataset, which depends on the downstream task rather than number of nodes in the graph. Specifically, suppose $\lambda_r$ is the $r$-th largest eigenvalue, an appropriate $r$ is chosen if there is a large gap between $\lambda_r$ and $\lambda_{r+1}$ (i.e., a large eigengap) in the graph spectrum. According to [39], the eigengap is highly related to the number of clusters in the graph. In this work, we approximate $r$ by $r \approx 10c$ to cover the large eigengap, where $c$ denotes the number of classes/clusters. As shown in Tables 1 and 6, the number of classes in small (large) graphs is around 5 (50), so we use $r = 50$ ($r = 500$) in experiments. As a result, GARNET has the near-linear time complexity $O(r(|E| + k|V|) + |V|log|V|) = O(c(|E| + k|V|) + |V|log|V|)$.

### F.2   Space Complexity

GARNET involves forming a sparse kNN graph by building hierarchical navigable small world (HNSW) graphs [28] that contain $O(|V|\log|V|)$ nodes in total and each node connects to a fixed number of neighbors. Thus, the space complexity of storing the HNSW graphs is $O(|V|\log|V|)$. In addition, GARNET also needs to store the input adversarial graph and the produced kNN graph. As a result, the total space complexity of GARNET is $O(|V|(\log|V| + k) + |\mathcal{E}|)$, where $|\mathcal{V}|$ and $|\mathcal{E}|$ denote the number of nodes and edges in the adversarial graph, respectively, and $k$ is the averaged node degree in the kNN graph.

Apart from the complexity analysis, we further provide the algorithms of GARNET and TSVD below for comparison. For GARNET algorithm, the embedding matrix $V$ at line 3 can be further concatenated with the node feature matrix, which may improve the quality of $\mathcal{G}_{base}$ as discussed in Appendix L. Moreover, lines 5 and 6 are optional as illustrated in Appendix S.

---

**Algorithm 1:** GARNET based adversarial defense (this work)

**Input:** Adversarial graph $\mathcal{G}_{adv}$; node feature matrix $X \in R^{n \times d}$; prior data variance $\sigma^2$; truncated svd rank $r$; kNN graph $k$; threshold for edge pruning $\gamma$; a GNN model for defense.

**Output:** Node embedding matrix $Z \in R^{n \times c}$

1   $M, S = eigs(\mathcal{G}_{adv}, r)$;

2   $V = M\sqrt{|I - S|}$;

3   $V = concat(V, X)$;                           ▷ optional

4   $\mathcal{G}_{base} = kNN\_graph(V, k)$;

5   $M', S' = eigs(\mathcal{G}_{base}, r)$;                ▷ optional

6   $U = M'/\sqrt{S' + I/\sigma^2}$;               ▷ optional

7   **for** $e_{i,j} \in \mathcal{G}_{base}$ **do**

8      **if** $\frac{\|U_i - U_j\|_2^2}{\|V_i - V_j\|_2^2} < \gamma$ *(or simplified version:* $\|V_i - V_j\|_2^2 > \gamma'$*)* **then**

9         Prune $e_{i,j}$ from $\mathcal{G}_{base}$;

10     **end**

11 **end**

12 $Z = \text{GNN}(\mathcal{G}'_{base}, X)$;

---

**Algorithm 2:** Truncated SVD based adversarial defense (prior work)

**Input:** Adversarial graph $\mathcal{G}_{adv}$; node feature matrix $X \in R^{n \times d}$; truncated svd rank $r$; a GNN model for defense.

**Output:** Node embedding matrix $Z \in R^{n \times c}$

13 $U, S, V = TSVD(\mathcal{G}_{adv}, r)$;

14 $A_{\mathcal{G}_{tsvd}} = USV^T$;

15 $Z = \text{GNN}(\mathcal{G}_{tsvd}, X)$;

---

# G   Ablation Study

### G.1   Choice of $k$ for kNN Graph Construction

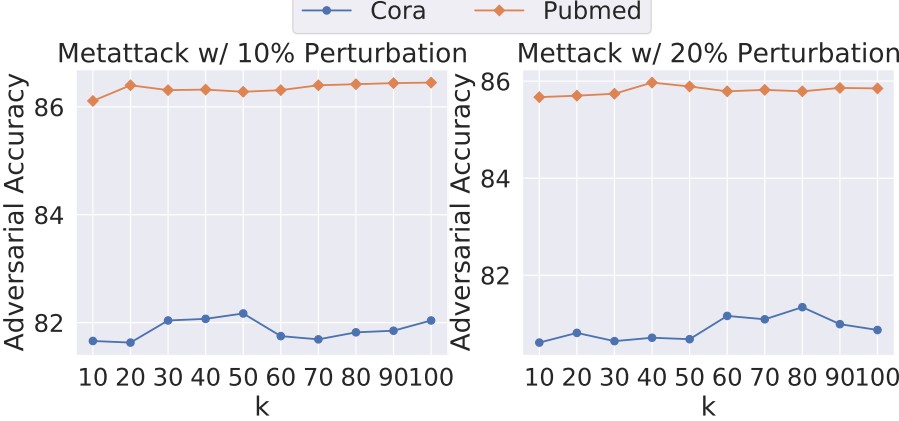

**Figure 6:** Ablation study of GARNET on $k$ for kNN graph construction.

To evaluate the sensitivity of GARNET to k nearest-neighbor (kNN) graph construction, we evaluate the adversarial accuracy of GARNET with different $k$ values for constructing kNN graphs. Figure

6 shows that the accuracy of GARNET does not change too much when varying $k$ value within the range of $[10, 100]$, indicating a relatively large $k$ (e.g., $k \geq 10$) can enable the kNN graph to incorporate most of edges in the underlying clean graph. Consequently, the performance of GARNET is relatively robust to the choice of $k$ for kNN graph construction. As the peak performance is typically achieved in $[30, 80]$, we recommend choosing $k = 30 \sim 80$ for building the kNN graph in practice.

### G.2 Choice of $\gamma$ for edge pruning

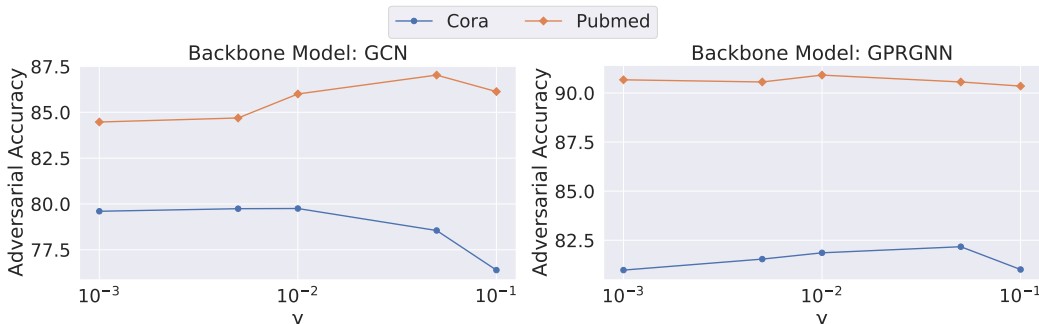

**Figure 7:** Ablation study of GARNET on threshold $\gamma$ for edge pruning.

Apart from the choice of $k$ for kNN graph construction, another critical hyperparameter of GARNET is the threshold $\gamma$ that determines whether an edge should be pruned in the base graph. Thus, we further evaluate the effect of $\gamma$ on the performance of GARNET. Specifically, we pick $\gamma$ in the set of $\{0.001, 0.005, 0.01, 0.05, 0.1\}$ and evaluate the corresponding adversarial accuracy of GARNET under Nettack with 1 perturbation per target node. As shown in Figure 7, the improper choice of $\gamma$ may degrade the adversarial accuracy of GARNET by 3%. It is worth noting that the proper value of $\gamma$ depends on the specific method adopted for measuring spectral embedding distortion. In other words, the optimal $\gamma$ will be different if we use the simplified version of spectral embedding distortion discussed in Appendix S. Thus, we refer readers to our official configuration files at github.com/cornell-zhang/GARNET/configs for the proper choices of $\gamma$ based on the simplified distortion metric.

## H  Backbone GNN Models for Defense

As GARNET can be integrated with any existing GNN models to improve their adversarial accuracy, we choose two popular GNN models as the backbone model in our experiments: GCN and GPRGNN [30, 31]. As the GCN model implicitly assumes the underlying graph is homophilic, it performs poorly on heterophilic graphs [10]. In contrast, GPRGNN can work on both homophilic and heterophilic datasets, due to its learned graph filter that can adapt to the homophily/heterophily property of the underlying graph. Thus, we choose GPRGNN as the backbone model for evaluation on heterophilic datasets. In addition, we also show the defense results with the H2GCN [10] as backbone model in Appendix J.

## I  Defense Results with Various Perturbation Budgets

We provide additional defense results under Nettack and Metattack with various perturbations in Tables 7 and 8 respectively. The results indicate that GARNET outperforms prior defense methods in most cases.

## J  Defense on H2GCN

We provide the results of combining GARNET with H2GCN [10] on heterophilic graphs in Table 9, which shows that GARNET achieves the highest accuracy in most cases and improves the accuracy on all perturbed graphs by a large margin compared to the vanilla H2GCN as well as H2GCN-TSVD. The results further confirm that GARNET is able to improve robustness of different backbone GNN models.

**Table 7:** Averaged node classification accuracy (%) ± std under targeted attack (Nettack) with different perturbation ratio — We denote the evaluated dataset by its name with the number of perturbations (e.g., Cora-0 means the clean Cora graph and Cora-1 denotes there is 1 adversarial edge perturbation per target node). As GCN is not designed for heterophilic graphs, we only show results of defense methods with GPRGNN as the backbone model on Chameleon and Squirrel. We bold and underline the first and second highest accuracy of each backbone GNN model, respectively. $OOM$ means out of memory.

| Dataset | GCN | | | | GPRGNN | | | |
|---|---|---|---|---|---|---|---|---|
| | Vanilla | TSVD | ProGNN | GARNET | Vanilla | TSVD | ProGNN | GARNET |
| Cora-0 | $\underline{80.96} \pm 0.95$ | $72.65 \pm 2.29$ | $80.54 \pm 1.21$ | $\mathbf{81.08} \pm 2.05$ | $\mathbf{83.04} \pm 2.05$ | $81.68 \pm 1.78$ | $82.04 \pm 1.33$ | $\underline{82.77} \pm 1.89$ |
| Cora-1 | $70.06 \pm 0.81$ | $71.36 \pm 1.63$ | $\mathbf{81.65} \pm 0.59$ | $\underline{79.75} \pm 2.35$ | $\underline{81.68} \pm 2.18$ | $79.36 \pm 2.23$ | $80.56 \pm 1.71$ | $\mathbf{82.17} \pm 1.95$ |
| Cora-2 | $68.60 \pm 1.81$ | $70.66 \pm 2.76$ | $\mathbf{79.83} \pm 1.10$ | $\underline{79.69} \pm 1.50$ | $74.34 \pm 2.41$ | $\underline{76.26} \pm 2.34$ | $76.12 \pm 2.43$ | $\mathbf{78.55} \pm 2.11$ |
| Cora-3 | $65.04 \pm 3.31$ | $68.20 \pm 1.93$ | $\underline{72.08} \pm 1.20$ | $\mathbf{74.42} \pm 2.06$ | $70.96 \pm 2.00$ | $70.90 \pm 3.89$ | $\underline{73.74} \pm 2.73$ | $\mathbf{79.40} \pm 1.35$ |
| Cora-4 | $61.69 \pm 1.48$ | $65.34 \pm 3.46$ | $\underline{67.83} \pm 1.87$ | $\mathbf{69.60} \pm 2.67$ | $65.90 \pm 1.61$ | $65.51 \pm 3.27$ | $\underline{68.94} \pm 3.25$ | $\mathbf{72.77} \pm 2.16$ |
| Cora-5 | $55.66 \pm 1.95$ | $60.30 \pm 2.25$ | $\underline{65.38} \pm 1.65$ | $\mathbf{67.04} \pm 2.05$ | $62.89 \pm 1.95$ | $63.52 \pm 3.27$ | $\underline{63.74} \pm 2.57$ | $\mathbf{71.45} \pm 2.73$ |
| Pubmed-0 | $87.26 \pm 0.51$ | $87.03 \pm 0.48$ | $\mathbf{88.14} \pm 1.44$ | $\underline{87.96} \pm 0.58$ | $\underline{90.05} \pm 0.73$ | $OOM$ | $OOM$ | $\mathbf{90.99} \pm 0.52$ |
| Pubmed-1 | $84.29 \pm 0.68$ | $\underline{86.46} \pm 0.28$ | $85.75 \pm 1.23$ | $\mathbf{87.03} \pm 0.68$ | $\underline{89.30} \pm 0.54$ | $OOM$ | $OOM$ | $\mathbf{90.91} \pm 0.47$ |
| Pubmed-2 | $82.17 \pm 0.67$ | $\underline{83.68} \pm 0.46$ | $81.23 \pm 1.21$ | $\mathbf{86.92} \pm 0.42$ | $\underline{87.42} \pm 0.28$ | $OOM$ | $OOM$ | $\mathbf{90.75} \pm 0.55$ |
| Pubmed-3 | $81.13 \pm 0.53$ | $\underline{81.34} \pm 0.68$ | $80.65 \pm 1.39$ | $\mathbf{86.50} \pm 0.45$ | $\underline{84.46} \pm 0.53$ | $OOM$ | $OOM$ | $\mathbf{90.70} \pm 0.37$ |
| Pubmed-4 | $75.48 \pm 0.52$ | $\underline{82.41} \pm 0.54$ | $78.46 \pm 1.11$ | $\mathbf{86.44} \pm 0.64$ | $\underline{81.72} \pm 0.72$ | $OOM$ | $OOM$ | $\mathbf{90.11} \pm 0.57$ |
| Pubmed-5 | $66.67 \pm 1.34$ | $\underline{79.56} \pm 0.48$ | $71.89 \pm 1.56$ | $\mathbf{86.12} \pm 0.86$ | $\underline{76.99} \pm 1.16$ | $OOM$ | $OOM$ | $\mathbf{89.52} \pm 0.45$ |
| Chameleon-0 | | | | | $\underline{71.46} \pm 1.92$ | $62.12 \pm 3.04$ | $58.80 \pm 1.72$ | $\mathbf{72.89} \pm 2.65$ |
| Chameleon-1 | | | | | $\underline{71.02} \pm 1.57$ | $61.34 \pm 2.93$ | $58.05 \pm 1.90$ | $\mathbf{72.68} \pm 1.89$ |
| Chameleon-2 | | | | | $\underline{70.71} \pm 1.12$ | $61.09 \pm 2.80$ | $57.44 \pm 1.67$ | $\mathbf{72.20} \pm 2.31$ |
| Chameleon-3 | | | | | $\underline{70.30} \pm 1.28$ | $60.98 \pm 2.82$ | $57.19 \pm 1.83$ | $\mathbf{72.17} \pm 2.07$ |
| Chameleon-4 | | | | | $\underline{69.87} \pm 1.29$ | $60.85 \pm 3.31$ | $57.44 \pm 1.63$ | $\mathbf{72.06} \pm 2.94$ |
| Chameleon-5 | | | | | $\underline{66.26} \pm 1.71$ | $60.37 \pm 2.86$ | $57.07 \pm 1.82$ | $\mathbf{71.83} \pm 2.11$ |
| Squirrel-0 | | | | | $\underline{41.36} \pm 2.87$ | $32.98 \pm 2.36$ | $31.81 \pm 1.72$ | $\mathbf{44.91} \pm 1.53$ |
| Squirrel-1 | | | | | $\underline{41.27} \pm 3.16$ | $32.63 \pm 0.87$ | $30.54 \pm 2.45$ | $\mathbf{43.55} \pm 1.79$ |
| Squirrel-2 | | | | | $\underline{41.09} \pm 2.14$ | $32.05 \pm 1.05$ | $30.73 \pm 2.13$ | $\mathbf{44.09} \pm 2.35$ |
| Squirrel-3 | | | | | $\underline{40.98} \pm 2.72$ | $32.00 \pm 1.66$ | $30.25 \pm 1.98$ | $\mathbf{44.18} \pm 2.26$ |
| Squirrel-4 | | | | | $\underline{40.25} \pm 2.82$ | $31.45 \pm 1.38$ | $29.09 \pm 2.33$ | $\mathbf{43.73} \pm 1.62$ |
| Squirrel-5 | | | | | $\underline{39.45} \pm 2.36$ | $31.20 \pm 1.84$ | $27.27 \pm 1.87$ | $\mathbf{43.64} \pm 1.53$ |

**Table 8:** Averaged node classification accuracy (%) ± std under non-targeted attack (Metattack) with different perturbation ratio — We denote the evaluated dataset by its name with the perturbation ratio (e.g., Cora-0 means the clean Cora graph and Cora-10 denotes there are 10% adversarial edges). As GCN is not designed for heterophilic graphs, we only show results of defense methods with GPRGNN as the backbone model on Chameleon and Squirrel. We bold and underline the first and second highest accuracy of each backbone GNN model, respectively. $OOM$ means out of memory.

| Dataset | GCN | | | | GPRGNN | | | |
|---|---|---|---|---|---|---|---|---|
| | Vanilla | TSVD | ProGNN | GARNET | Vanilla | TSVD | ProGNN | GARNET |
| Cora-0 | $\mathbf{81.35} \pm 0.66$ | $73.86 \pm 0.53$ | $78.56 \pm 0.36$ | $\underline{79.64} \pm 0.75$ | $\mathbf{83.05} \pm 0.42$ | $81.61 \pm 0.54$ | $82.04 \pm 0.90$ | $\underline{82.67} \pm 1.89$ |
| Cora-10 | $69.50 \pm 1.46$ | $69.45 \pm 0.69$ | $\mathbf{77.90} \pm 0.69$ | $\underline{77.78} \pm 0.53$ | $80.37 \pm 0.65$ | $\underline{81.08} \pm 0.52$ | $80.31 \pm 1.23$ | $\mathbf{82.17} \pm 0.69$ |
| Cora-20 | $56.28 \pm 1.19$ | $62.44 \pm 1.16$ | $\underline{72.28} \pm 1.67$ | $\mathbf{73.89} \pm 0.91$ | $74.27 \pm 2.11$ | $\underline{78.50} \pm 1.20$ | $76.29 \pm 1.46$ | $\mathbf{81.34} \pm 0.79$ |
| Pubmed-0 | $\mathbf{87.16} \pm 0.09$ | $84.53 \pm 0.08$ | $84.62 \pm 0.11$ | $\underline{85.37} \pm 0.20$ | $\mathbf{87.35} \pm 0.13$ | $OOM$ | $OOM$ | $\underline{86.86} \pm 0.57$ |
| Pubmed-10 | $81.16 \pm 0.13$ | $\underline{84.56} \pm 0.10$ | $84.09 \pm 0.12$ | $\mathbf{85.22} \pm 0.13$ | $\underline{85.52} \pm 0.14$ | $OOM$ | $OOM$ | $\mathbf{86.24} \pm 0.20$ |
| Pubmed-20 | $77.20 \pm 0.27$ | $\underline{84.30} \pm 0.08$ | $83.89 \pm 0.32$ | $\mathbf{85.14} \pm 0.23$ | $\underline{84.18} \pm 0.15$ | $OOM$ | $OOM$ | $\mathbf{85.69} \pm 0.26$ |
| Chameleon-0 | | | | | $\mathbf{61.36} \pm 1.00$ | $47.29 \pm 1.63$ | $48.39 \pm 0.68$ | $\underline{61.11} \pm 2.46$ |
| Chameleon-10 | | | | | $\underline{57.55} \pm 1.26$ | $47.07 \pm 1.21$ | $47.80 \pm 0.91$ | $\mathbf{60.96} \pm 1.22$ |
| Chameleon-20 | | | | | $\underline{53.20} \pm 0.88$ | $45.12 \pm 1.34$ | $46.69 \pm 0.61$ | $\mathbf{59.96} \pm 0.84$ |
| Squirrel-0 | | | | | $\underline{39.51} \pm 1.64$ | $31.36 \pm 1.87$ | $31.64 \pm 2.87$ | $\mathbf{43.43} \pm 1.14$ |
| Squirrel-10 | | | | | $\underline{38.27} \pm 0.83$ | $28.25 \pm 1.66$ | $30.33 \pm 3.29$ | $\mathbf{42.62} \pm 1.09$ |
| Squirrel-20 | | | | | $\underline{35.22} \pm 1.20$ | $23.91 \pm 1.40$ | $29.36 \pm 3.61$ | $\mathbf{41.97} \pm 1.02$ |

**Table 9:** Averaged node classification accuracy (%) ± std on heterophilic graphs — We bold and underline the first and second highest accuracy, respectively. The backbone GNN model is H2GCN.

| Model | Chameleon (Nettack) | | Chameleon (Metattack) | | Squirrel (Nettack) | | Squirrel (Metattack) | |
|---|---|---|---|---|---|---|---|---|
| | Clean | Adversarial | Clean | Adversarial | Clean | Adversarial | Clean | Adversarial |
| Vanilla | $\underline{78.43} \pm 2.09$ | $62.20 \pm 1.99$ | $\mathbf{68.45} \pm 0.57$ | $52.73 \pm 1.72$ | $\mathbf{55.36} \pm 2.91$ | $29.55 \pm 3.09$ | $\mathbf{61.23} \pm 0.71$ | $\underline{44.84} \pm 0.89$ |
| TSVD | $67.07 \pm 1.15$ | $\underline{63.17} \pm 1.61$ | $61.75 \pm 1.09$ | $\underline{54.06} \pm 1.66$ | $32.45 \pm 1.87$ | $\underline{31.64} \pm 2.09$ | $46.66 \pm 1.71$ | $40.56 \pm 1.41$ |
| GARNET | $\mathbf{78.78} \pm 1.84$ | $\mathbf{76.10} \pm 1.92$ | $\underline{66.63} \pm 1.05$ | $\mathbf{61.12} \pm 0.59$ | $\underline{54.09} \pm 1.73$ | $\mathbf{53.27} \pm 1.50$ | $\underline{59.67} \pm 0.83$ | $\mathbf{50.08} \pm 1.92$ |

# K    Broader Impact

Zügner et al. [5] have shown that graph adversarial attacks can drastically degrade the performance of GNN models for downstream applications. For instance, an attacker can attack a GNN-based recommender system on Facebook social network or Amazon co-purchasing network, via creating a fake account and make some connections to other users or items. Those connections can be viewed as adversarial edges in the graph. As a result, the attacker can deliberately enforce a GNN model to recommend some irrelevant or even harmful contents to other users. Thus, improving adversarial robustness of GNN models has the potential for positive societal benefit.

We hope that this paper provides insight on the robustness and scalablity limitations of prior defense methods. Moreover, we believe that the proposed GARNET can largely overcome these two limitations and produce a robust GNN model against adversarial attacks on large-scale graph datasets. Nevertheless, we have to admit that GARNET may potentially provide the attacker with some hints about developing an even more powerful and scalable adversarial attack than all existing attacks, which is a possible negative consequence.

# L    Discussion on Node Features

## L.1    Graph Construction with Node Features

**Table 10:** Averaged node classification accuracy (%) $\pm$ std under targeted attack (Nettack) and non-targeted attack (Metattack) on Cora and Pubmed — We bold and underline the first and second highest accuracy, respectively. "NodeFeat" denotes the kNN graph constructed from node features is used for GNN training. "GARNET w/ NodeFeat" denotes the kNN graph constructed from the concatenation of dominant singular components and node features. The backbone model is GCN.

| Model | Cora (Nettack) | | Cora (Metattack) | | Pubmed (Nettack) | | Pubmed (Metattack) | |
|---|---|---|---|---|---|---|---|---|
| | Clean | Adversarial | Clean | Adversarial | Clean | Adversarial | Clean | Adversarial |
| Vanilla | $80.96 \pm 0.95$ | $55.66 \pm 1.95$ | $\underline{81.35} \pm 0.66$ | $56.28 \pm 1.19$ | $87.26 \pm 0.51$ | $66.67 \pm 1.34$ | $\mathbf{87.16} \pm 0.09$ | $77.20 \pm 0.27$ |
| NodeFeat | $52.65 \pm 2.69$ | $52.65 \pm 2.69$ | $56.44 \pm 1.04$ | $56.44 \pm 1.04$ | $83.01 \pm 0.99$ | $83.01 \pm 0.99$ | $78.66 \pm 0.15$ | $78.66 \pm 0.15$ |
| GARNET | $\underline{81.08} \pm 2.05$ | $\mathbf{67.04} \pm 2.05$ | $79.64 \pm 0.75$ | $\mathbf{73.89} \pm 0.91$ | $\underline{87.96} \pm 0.58$ | $\underline{86.12} \pm 0.86$ | $85.37 \pm 0.20$ | $\mathbf{85.14} \pm 0.23$ |
| GARNET w/ NodeFeat | $\mathbf{83.73} \pm 1.17$ | $\underline{64.35} \pm 2.98$ | $\mathbf{81.93} \pm 0.44$ | $\underline{71.97} \pm 0.95$ | $\mathbf{88.76} \pm 0.40$ | $\mathbf{86.56} \pm 0.62$ | $\underline{86.16} \pm 0.16$ | $84.88 \pm 0.34$ |

As GARNET purifies the adversarial graph by building a kNN graph based on dominant singular components, a natural question is whether the kNN graph constructed from node features can also achieve similar performance. We answer this question by comparing the results of GARNET graph and the node feature graph in Table 10. Note that the clean and adversarial accuracy are the same on the graph constructed from node features, since node features are unchanged after graph adversarial attack. Besides, we only show results on homophilic graphs as the kNN graph constructed from node features naturally falls into this category. Table 10 shows that the node feature graph performs much worse than GARNET graph. This further confirms that the method proposed in this work is critical to improve the robustness of GNN models.

Apart from constructing the kNN graph purely from node features, we can also concatenate node features with dominant singular components for kNN graph construction, which may further improve the accuracy of GARNET. Note that we only adopt this notion for homophilic graphs, as this approach implicitly assumes that the graph is homophilic (nodes with similar features are adjacent in the kNN graph). The results in Table 10 indicate that augmenting GARNET with node features can further improve the accuracy in several cases.

## L.2    Defense Against Node Feature Attack

GARNET can be extended to handle node feature attack, although this paper mainly focuses on defending against graph structure attack, which we believe is more challenging than defending node feature attack due to the discrete nature. Specifically, we can perform TSVD to obtain the low-rank approximation of the node feature matrix, which can remove high-rank adversarial components in node features [7]. The low-rank feature matrix is then concatenated to the weighted spectral embeddings to produce the kNN base graph. In this way, the downstream GNN model will be able to aggregate neighbors whose features are less perturbed during message passing.

# M  Additional Visualization Results

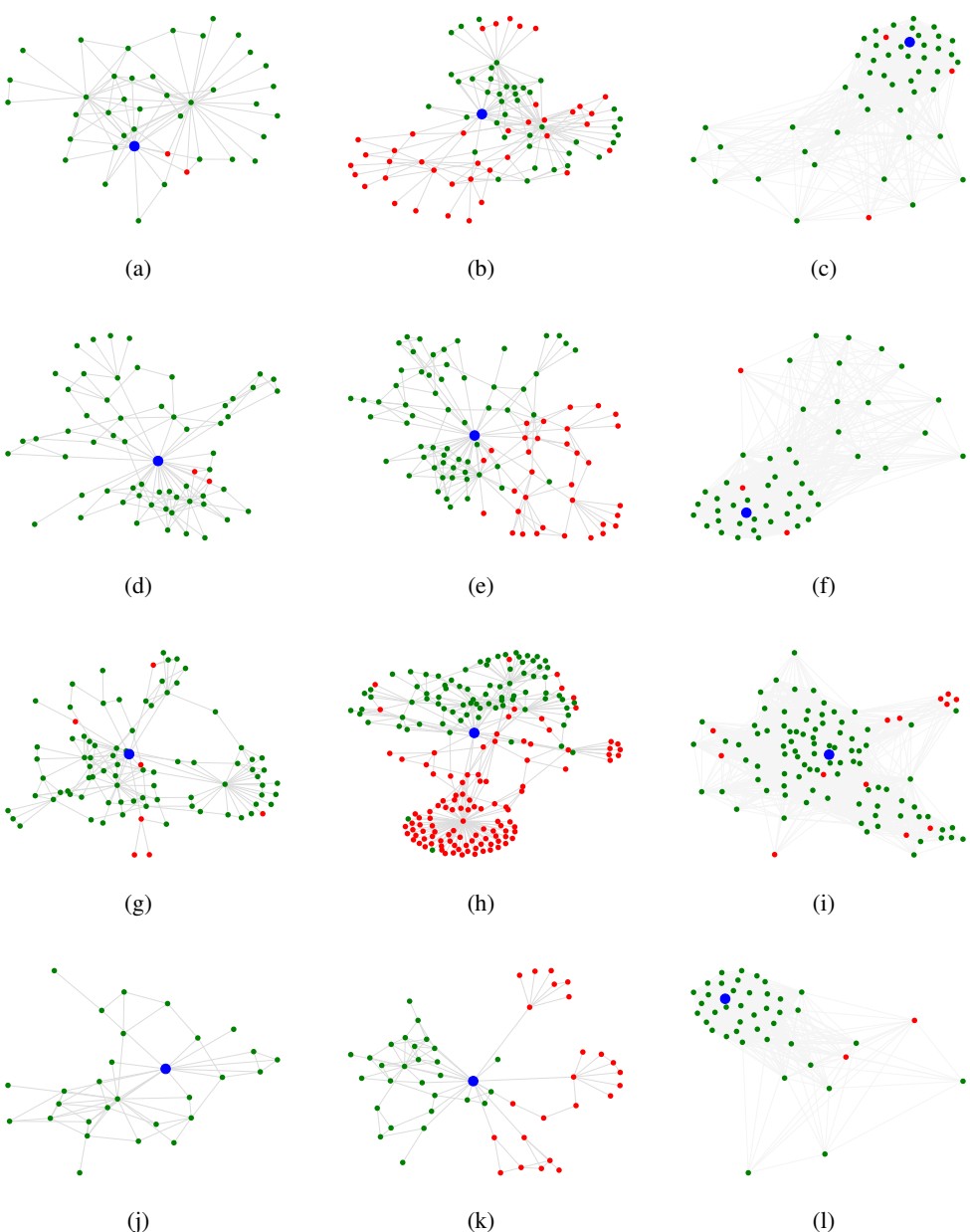

(a)                    (b)                    (c)

(d)                    (e)                    (f)

(g)                    (h)                    (i)

(j)                    (k)                    (l)

**Figure 8:** Cora visualizations on a target node (marked in blue) as well as its 1-hop and 2-hop neighbors. Neighbor nodes are marked in green if they have the same label as the target node, and red otherwise. Note that the three graphs in the same row share the same target node (randomly picked), while graphs in different rows focus on different target nodes. **Left**: clean graph. **Middle**: adversarial graph. **Right**: adversarial graph purified by GARNET.

We visualize more target nodes and their local structures in Figure 8, which reveals that GARNET consistently improves the quality of adversarial graph by removing adversarial edges that connect nodes with different labels. As a result, the adversarial accuracy of backbone GNN models can be largely improved once they are trained on the GARNET graph.

## N  Acceleration of GARNET on Large Graphs

There are two major kernels in GARNET: (1) weighted spectral embedding (i.e., computing the top $r$ singular components or Laplacian eigenpairs), (2) kNN graph construction. For accelerating weighted spectral embedding, we can leverage the notion of multi-level graph coarsening [40, 41] so that we only need to perform TSVD on the coarsest graph. To speedup the process of kNN graph construction, we can exploit Faiss to enable performing kNN on GPU [42].

## O  Homophily Score of GARNET Graph

**Table 11:** Graph homophily score.

|  | Homophilic graphs | | Heterophilic graphs | |
| --- | --- | --- | --- | --- |
| Dataset | Cora | Pubmed | Chameleon | Squirrel |
| Clean graph | 0.80 | 0.80 | 0.23 | 0.22 |
| GARNET graph | 0.75 | 0.72 | 0.25 | 0.26 |

We follow Zhu et al. [10] to compute the homophily score per dataset (lower score means more heterophilic). As shown in Table 11, the GARNET graph is homophilic (heterophilic) if the corresponding clean graph is homophilic (heterophilic), which further confirms Theorem 3.3 that our approach can effectively recover the clean graph structure. As a result, GARNET supports both homophilic and heterophilic graphs.

## P  Accuracy of Clean Graph Recovery

**Table 12:** Averaged recall and precision of clean structure recovery over 5 (randomly picked) nodes.

|  | Recall | Precision |
| --- | --- | --- |
| Cora (homophilic graph) | 0.94 | 0.65 |
| Chameleon (heterophilic graph) | 0.87 | 0.59 |

Apart from visualizing GARNET graph in Figures 5 and 8, we further quantify how well GARNET recovers the clean graph structure. Concretely, given a target node, we first extract nodes within its 2-hop neighbors in the clean graph and GARNET graph (under Metattack with $20\%$ perturbation ratio), respectively. By denoting the extracted nodes by $N_{clean}$ for clean graph and $N_{garnet}$ for GARNET graph, we define the recall score and precision score as follows:

$$Recall = \frac{|N_{clean} \cap N_{garnet}|}{|N_{clean}|}$$

$$Precision = \frac{|N_{clean} \cap N_{garnet}|}{|N_{garnet}|}$$

Table 12 shows the averaged recall and precision over 5 nodes on Cora and Chameleon graphs. The results show that the recall scores are very high for both graphs, which indicates GARNET is able to accurately recover clean graph structure. The relatively low precision scores indicate that GARNET also introduces new edges to the graph (i.e., $|N_{garnet}| > |N_{clean}|$). We argue that those new edges are likely to connect spectrally similar nodes that are far away in the original clean graph, which enables GARNET to also incorporate global structural information. This explains why GARNET can sometimes outperform vanilla GNN models on clean heterophilic graphs (shown in Table 3), where global structural information is very critical for node prediction.

## Q    Further Discussion on Graph Recovery with PGM

Intuitively, if we use more (clean) Laplacian eigenpairs (i.e., a larger $r$) for constructing the embedding matrix $V$ based on Definition 3.1, the optimal solution for Equation 2 (i.e., $\Theta^*$) will be closer to the actual clean graph. In this section, we confirm this intuition based on graph resistance distances between node pairs. Specifically, consider the following expression for calculating effective-resistance distances between nodes $p$ and $q$ using all Laplacian eigenvalues/eigenvectors except $\lambda_1 = 0$:

$$\sum_{i=2}^{|V|} \frac{(u_i^T e_{p,q})^2}{\lambda_i}$$

Feng [17] has shown that the effective-resistance distance between any node pair on the learned graph $\Theta^*$ (when $\sigma$ approaches infinity) will fully match the Euclidean distance between the corresponding data samples (i.e., rows in weighted spectral embedding matrix $V$ in our case). Moreover, it can be shown that the Euclidean distance between the data samples in our case will match the effective-resistance distance on the original graph when $r = |\mathcal{V}|$ (with proper normalization on Laplacian eigenvectors). As a result, the resistance distances on the learned graph $\Theta^*$ will be the same as the ones on the original graph when $r = |\mathcal{V}|$. Moreover, using a larger $r$ value will lead to a more accurate estimation of the learned (clean) graph.

In practice, if $r$ satisfies that $\lambda_r \ll \lambda_{r+1}$, dropping the terms with much larger eigenvalues (i.e., $\lambda_{r+1}$, $\lambda_{r+2}$, ..., $\lambda_{|\mathcal{V}|}$) will not significantly impact the approximation accuracy. A proper $r$ can be effectively determined based on the strategy proposed in Appendix F. We leave the theoretical guarantee of other metrics for graph comparison to our future work.

## R    Connection Between kNN Graph and TSVD Graph

Apart from the motivation of constructing a kNN graph as $\mathcal{G}_{base}$ based on Theorem 3.3, we further motivate the kNN graph construction from the perspective of improving the scalability of TSVD-based defense methods. Concretely, as previous TSVD-based methods produce a dense (low-rank) adjacency matrix $\hat{A}$, they involve dense matrices during GNN training, which has quadratic time/space complexity and thus cannot scale to large graphs. A potential solution is to sparsify $\hat{A}$ by preserving the top $k$ largest elements per row. However, naïvely selecting the largest elements of each row in $\hat{A}$ requires forming/storing $\hat{A}$ first, which still has quadratic time/space complexity. In contrast, we leverage the (approximate) kNN algorithm to construct the sparsified $\hat{A}$ by taking as input the weighted spectral embedding $V$ (note that $\hat{A} = VV^T$ based on Proposition 3.2). Consequently, our kNN graph construction step can also be viewed as a scalable way of sparsifying the dense adjacency matrix $\hat{A}$ generated by TSVD. Moreover, Theorem 3.3 theoretically guarantees that the sparsified graph serves as a reasonable $\mathcal{G}_{base}$ for edge pruning.

## S    Simplified Version of Spectral Embedding Distortion

To obtain $s_{i,j} = \frac{\|U^T e_{i,j}\|_2^2}{\|V^T e_{i,j}\|_2^2}$, we have to compute both the top $r$ Laplacian eigenpairs of $\mathcal{G}_{base}$ (for constructing $U$) and those of $\mathcal{G}_{adv}$ (for constructing $V$). However, as $\mathcal{G}_{base}$ is relatively dense (a large $k$ used for kNN graph construction), computing its Laplacian eigenpairs is time-consuming, especially on large graphs. Fortunately, Von Luxburg [39] has shown that the top $r$ Laplacian eigenvectors (corresponding to smallest eigenvalues) only vary a little across nodes in a dense graph (i.e., they are smooth over the graph), which means $\|U^T e_{i,j}\|_2$ is very similar across different edges in $\mathcal{G}_{base}$. Consequently, the term $\|V^T e_{i,j}\|_2$ becomes the dominant factor in $s_{i,j}$ to identify (non)critical edges: a large value of $\|V^T e_{i,j}\|_2$ means a small value of $s_{i,j}$, indicating the corresponding edge is noncritical. As we empirically find out that exploiting $\|V^T e_{i,j}\|_2$ to prune noncritical edges does not degrade the accuracy of GARNET, we adopt this simplified version of spectral embedding distortion in our implementation.

