# OpenReview forum: "GARNET: Reduced-Rank Topology Learning for Robust and Scalable Graph Neural Networks"
_logconference.io/LOG/2022/Conference — LoG 2022 Oral_

### Official Review · Reviewer_SBPS · 2022-10-20

**Overall Score:** 8
**Confidence:** 4

**Review:**

Summary:

This work investigates how to defend against adversarial attacks on graphs, especially on large-scale ones. The authors propose GARNET, which is a scalable spectral model. GARNET is not only robust against adversarial attacks but also learn from clean structures for better performance.

Pros:
- This paper is clearly motivated and easy to follow.
- The methodological designs are novel, intuitive, and theoretically sound.
- The experimental results are extensive for both attacking algorithms, baselines defenses, backbone models, and datasets. The numerical results are strong.
- End-to-end training speed of GARNET wins by a large margin, showing the scalability of the method. Large-scale datasets are used.

Cons:
- I don't see outstanding drawback of this work.

Questions:

In Table 1 & 2, there are cases where advesarially trained backbones attain better results even than the vanilla model. It is well-known that adversarially robust models are prone to have worse clean accuracy. Do we have understanding towards this situation? Is there any chance that GARNET can serve to generalize GNN performance on clean graphs?

Recommendation:
Accept. Because of the novelty and soundness of the method as well as the strong experimental results, I vote for a clear accept.

---

### Official Review · Reviewer_v1A2 · 2022-10-22

**Overall Score:** 8
**Confidence:** 4

**Review:**

## Summary

This paper proposed GARNET, a scalable GNN model with boosted robustness against adversarial structural attacks of GNNs by recovering the clean graph structure using similarity of low-rank spectral embeddings and edge pruning on probabilistic graphical model; the recovered graph has a slightly reduced rank compared to the clean and perturbed graph, but has significantly higher ranks compares to the low-rank graphs adopted in prior defense approaches. The proposed approach is justified by theoretical and extensive empirical evaluations, which includes a wide range of baseline approaches and datasets with large scales or heterophily; the proposed approach significantly increases the robustness for multiple base GNNs on the aforementioned datasets.

## Overall Recommendation

**I recommend acceptance of this paper**, as it is clear based on the strengths listed below that this is a solid work, and its publication will benefit the robustness GNN research community. I hope the authors will further address the limitations and questions listed below.

## Strengths

1. **This paper proposed a novel approach that addressed several drawbacks of existing approaches for improving GNN robustness.**
2. **The effectiveness of the proposed approach is demonstrated in comprehensive experiments** that consider a wide range of baseline approaches (TSVD, ProGNN, GNNGuard, GCN-LFR), base GNN models (GCN, GPRGNN, and H2GCN), and datasets with varying levels of homophily.
3. **The proposed robustified GNN demonstrates good scalability to larger graph datasets with 2 million nodes and 61.8 million edges (ogbn-products).** This is worth applauding as prior works in improving GNN robustness largely ignore the scalability issue, which is important for applicability in large-scale datasets that are more commonly seen in the industry.
4. **Theoretical justifications and ablation studies are provided to justify the design choices of the proposed approach.**
5. **The writing of the paper is overall clear and easy to follow.**

## Limitations & Weaknesses

1. **The runtime comparison in Section 4.2 is limited to GNNs with improved robustness.** It would be good to include the runtimes of vanilla GNNs without improved robustness to demonstrate the trade-offs between robustness and speed.
2. **It is still not fully clear to me how attractive Gaussian Markov random fields (GMRFs)** 1. **help identify edges for pruning.** Providing further intuitions with examples or illustrations may help the reader to better understand the mechanism behind the hard-to-follow equations.

## Questions for Authors

1. **How do the edges in GARNET-recovered graphs overlap with the edges in the clean graph?** Though Section 4.4 visualizes and compares the level of homophily in 2-hop neighborhoods of randomly selected nodes in Cora between the GARNET-recovered graph and the original graph, providing precision and recall of edges on both homophilous and heterophilous dataset will bring more insights to the behavior of the proposed approach.

---

### Official Review · Reviewer_D9s8 · 2022-10-22

**Overall Score:** 8
**Confidence:** 5

**Review:**

### Paper Summary
This paper presents GARNET, a graph adversarial defense approach that recovers the underlying clean graph topology of an adversarial graph utilizing the truncated spectral eigenvalue & eigenvector pairs and probabilistic graphical model (PGM). GARNET first employs weighted spectral eigenvectors to build a k-nearest neighbor (kNN) base graph, which is subsequently pruned by removing non-critical edges based on an estimated Gaussian Markov random fields (GMRF) model. GARNET outperforms TSVD and ProGNN against the targeted attack (Nettack) and the non-targeted attack (Metattack) and scales to large graphs with millions of nodes, as demonstrated by the experiments.

### Strengths and Weaknesses
#### Strengths
1. The authors provide extensive empirical assessments of the proposed framework. The datasets include both homophilic and heterophilic graphs and range from small to large ones. The attack methods include both targeted and untargeted attacks. The authors also demonstrated the efficiency of the proposed method on small and large graphs (in Figure 3 and Appendix M). One small suggestion is to move the few numbers in Appendix M to the main paper, as the efficiency results on the large graph are more important.
2. I like the idea of pruning the base graph using an estimated Gaussian Markov random fields (GMRF) model and the simplicity of Theorem 3.3. It is easy to see the proposed method should be applied equally well to homophilic and heterophilic graphs.


#### Weaknesses
1. One issue is that the authors claim that the truncated singular value decomposition (TSVD) based defense methods have high time/space complexities and cannot scale to large graphs because "they involve dense adjacency matrices" (Line 33). Firstly, TSVD can be efficiently computed with the sparse matrix representation, e.g., see sklearn's implementation https://scikit-learn.org/stable/modules/generated/sklearn.decomposition.TruncatedSVD.html, or PyTorch's https://pytorch.org/docs/stable/generated/torch.pca_lowrank.html which both support sparse matrix. Secondly, if the low-rank adjacency matrix they computed (as the product of TSVD outputs) is dense (and may even have negative edge weights), could we consider a simple baseline, which filters out the negative edge weights, and sparsify the graph by simply selecting the $k$-largest weighted egdes for each node? Such a simple procedure would produce a sparse graph and is likely to be helpful to the performance as well. If time allows, I would like to see some ablation studies on this.
2. I find the description of Eq. (4) confusing and possibly incorrect (please correct me if I am wrong). In Line 248, it is said that "$\Theta$ is constrained to be $L+\frac{I}{\sigma^2}$" (where $L=D-A$ as described in Line 90. I find this sentence a bit confusing. I think by this constraint, the authors meant to say the positions of the (off-diagonal) non-zero entries of $\Theta$ are fixed by the given graph (e.g., $\mathcal{G}\_{base}$), but the edge weights are free to learn. In this regard, the degree matrix $D$ and the eigenvalues $\lambda_k$ in Eq. (4) should be affected by the edge weights $w_{i,j}$, right? So $\lambda_k$ in Eq. (4) should not be the Laplacian eigenvalues of the input kNN graph $\mathcal{G}\_{base}$ (which is unweighted) as described in Line 258, otherwise $\partial \lambda_k/\partial w_{i,j}=0$. Moreover, $\lambda_k$ should be a variable depending on $\Theta$ (otherwise, the right-hand side of Eq. (4) is a constant, and the partial derivative $\partial F/\partial w_{i,j}$ may never be zero.) Then why it is Ok to define $U$ and the spectral embedding distortion $s_{i,j}$ using just the eigenvalues of $\mathcal{G}_{base}$?
3. The clarity of some descriptions in section 3 needs some improvements (see the other issues listed below). And the related work section needs some more details (see the other question 1 & 2 listed below.)


### Other Questions
1. Did the TSVD method assume graphs to be homophilic as well? Could you briefly describe why ProGNN (and/or TSVD) defenses rely on the high-homophily assumption in section 2.3?
3. For Eq. (2), the authors claim that "we can recover the underlying graph that maximizes the likelihood." Is there any theoretical guarantee that if $r$ is large enough, the optimizer $\Theta^*$ will be close enough to the original graph $A$?


### Other Issues
1. I think you don't have to assume $\|V_i\|_2=1$, i.e., $V_i$ is normalized for all $i\in[n]$ in Theorem 3.3. This seems not used in the proof. And it contradicts your Definition 3.1. By Definition 3.1, easy to find that, any row $V_i$ can be constructed by (1) first finding the truncated vector (keeping the first $r\ll n$ element) of the corresponding $i$-th row of $Q=[v_1,\ldots,v_n]$ (denoted by $Q_i$, where $Q$ is an orthogonal matrix $QQ^T=I$), and (2) scaled down each element by multiplying with $\sqrt{|1-\lambda_r|}\leq1$. From the above process, easy to see that $\|V_i\|_2<\|Q_i\|_2=1$.
2. In Line 182, it is said that "To recover the underlying graph structure from the data matrix $X$"; however, what is actually proposed is to make use of the spectral "embedding" matrix $V$ (Eq. (2)). I suggest not to confuse these two concepts (i.e., feature v.s. adjacency), and not to call the matrix $V$ as an embedding matrix, because $V$ does not depends on the feature matrix $X$.
3. In Line 142, I think it is not appropriate to say "eliminating the high-rank components." Instead, it is more appropriate to say, "eliminating the high-frequency components of its spectrum." Also, please note that the eigenvalues of adjacency matrix $A$ are different from the spectrum, i.e., eigenvalues of graph Laplacian $L$. So the most rigorous way to say this is maybe just "finding the low-ranked approximations of the adjacency."

### Overall Recommendations
Overall I recommend weak acceptance for this current manuscript, and the major reasons are: (1) I think the scalability contribution is a bit overclaimed (weakness 1) and (2) the rigor and clarity of the theory/algorithm parts need some improvements (weakness 2, 3). I would like to consider raising my score if the authors can address my concerns in the rebuttal period.

### Edit during the rebuttal
Since the authors have addressed many of my concerns, I decided to raise my score to clear accept.

---

### Meta-Review · Area_Chair_mNCT · 2022-11-15

**Confidence:** 4
**Recommendation:** Accept

**Meta Review:**

The proposed method shows an effective strategy to train GNNs that are robust to different kinds of adversarial attacks. The proposed method has theoretical insights, as well as novel ideals (drawing connections with graphical models). The experiments are comprehensive. I would like to recommend acceptance.

---

### Decision · Program_Chairs · 2022-11-22

Accept (Oral)